# SOX11 and SOX4 drive the reactivation of an embryonic gene program during murine wound repair

Qi Miao[1,2], Matthew C. Hill [3], Fengju Chen[4], Qianxing Mo[4,5], Amy T. Ku[1,2,6], Carlos Ramos [1,2], Elisabeth Sock [7], Véronique Lefebvre[8] & Hoang Nguyen [1,2,3,4,6,9]

Tissue injury induces changes in cellular identity, but the underlying molecular mechanisms remain obscure. Here, we show that upon damage in a mouse model, epidermal cells at the wound edge convert to an embryonic-like state, altering particularly the cytoskeletal/extracellular matrix (ECM) components and differentiation program. We show that SOX11 and its closest relative SOX4 dictate embryonic epidermal state, regulating genes involved in epidermal development as well as cytoskeletal/ECM organization. Correspondingly, postnatal induction of SOX11 represses epidermal terminal differentiation while deficiency of *Sox11 and Sox4* accelerates differentiation and dramatically impairs cell motility and re-epithelialization. Amongst the embryonic genes reactivated at the wound edge, we identify fascin actin-bundling protein 1 (FSCN1) as a critical direct target of SOX11 and SOX4 regulating cell migration. Our study identifies the reactivated embryonic gene program during wound repair and demonstrates that SOX11 and SOX4 play a central role in this process.

[1] Stem Cells and Regenerative Medicine Center, Baylor College of Medicine, 1 Baylor Plaza, BCM 505, Houston, TX 77030, USA. [2] Center for Cell and Gene Therapy, Baylor College of Medicine, 1 Baylor Plaza, BCM 505, Houston, TX 77030, USA. [3] Program in Developmental Biology, Baylor College of Medicine, 1 Baylor Plaza, BCM 505, Houston, TX 77030, USA. [4] Dan L. Duncan Cancer Center, Baylor College of Medicine, 1 Baylor Plaza, BCM 505, Houston, TX 77030, USA. [5] Department of Biostatistics & Bioinformatics, H. Lee Moffitt Cancer Center, 12902 USF Magnolia Drive, Tampa, FL 33612, USA. [6] Interdepartmental Program in Translational Biology and Molecular Medicine, Baylor College of Medicine, 1 Baylor Plaza, BCM 505, Houston, TX 77030, USA. [7] Institut für Biochemie, Emil-Fischer-Zentrum, Friedrich-Alexander-Universität Erlangen-Nürnberg, Fahrstrasse 17, 91054 Erlangen, Germany. [8] Department of Surgery/Division of Orthopedic Surgery, Children's Hospital of Philadelphia, 3615 Civic Center Blvd, Philadelphia, PA 19104, USA. [9] Department of Molecular and Cellular Biology, Baylor College of Medicine, 1 Baylor Plaza, BCM 505, Houston, TX 77030, USA. Correspondence and requests for materials should be addressed to Q.M. (email: maioqi.res@gmail.com) or to H.N. (email: hoang.nguyenh@gmail.com)

The epidermis, the outermost layer of the skin, provides a protective barrier from the external environment[1]. Injuries disrupting the skin barrier elicit responses from multiple cell types, inducing epidermal cells to migrate to the wound site and regenerate a new epidermis[2]. During this wound-healing process, epidermal cells are reprogrammed to be more plastic in their fate, but the regulation of this process remains poorly defined[3,4].

Recent reports show that injuries induce the expression of a fetal-like transcriptional program in epithelial cells lining the digestive system[5–7]. The reactivation of certain embryonic genes is also observed in other regenerative processes such as bone fracture repair[8–10] and nerve regeneration[11–13]. Interestingly, in both nerve and bone regeneration the embryonic gene SOX11 is induced and implicated in the regenerative process[10–13]. The relationship between SOX11 and the reactivation of embryonic genes during tissue repair is unclear. Moreover, whether the reactivation of an embryonic gene program is required for regeneration and how the reactivation of this program is regulated remain to be elucidated.

SOX11 belongs to the *SRY*-related high-mobility-group (HMG) box family of transcription factors, classified into eight subgroups on the basis of the degree of conservation of the HMG DNA-binding domain and the surrounding sequences[14,15]. Part of the SOXC subfamily, *Sox11*, *Sox4*, and *Sox12* are crucial yet partially redundant during development[16–18]. *Sox11*-null mice die at birth, from multiple defects, including faulty heart development, craniofacial and skeletal malformations, and multiple hypoplastic organs[19]. *Sox4-null* mice die embryonically at E14.5, with severe heart defects and arrested B-cell differentiation[20]. Mice lacking both *Sox4* and *Sox11* die earlier, by E10.5, and have more severe developmental failure, highlighting the partial redundancy of *Sox11* and *Sox4*[17]. *Sox12*-null mice are normal and fertile[21], but the ablation of *Sox12* in mice already lacking *Sox4* and *Sox11* aggravates the developmental failure still further. *Sox12* thus plays a minor role in mouse development, with some overlapping functions with *Sox11* and *Sox4*[17]. The role of SOXC family members in skin is still unclear. Although *Sox4* has been ablated in skin, its precise role remains ambiguous due to the extreme hypomorphism of the line that contains the *loxP* insertion in the *Sox4* locus[22].

In addition to its role in embryonic development, SOX11 is induced in adult neuronal and mesenchymal cells upon injury and contributes to nerve regeneration and bone repair[10–13]. As SOX11 clearly holds a critical role in embryonic development and regeneration[10,11,13,17,19], we postulate that SOX11 might be an important molecular link between the embryonic state and tissue regeneration.

In this study, we explore the relationship between the reactivation of an embryonic gene program and wound repair. We show that one-quarter of all wound-induced epidermal signature genes[4] are embryonic epidermal signature genes and identify *Sox11* as one of the wound-induced embryonic transcription factors. We demonstrate that overexpressing *Sox11* in the epidermis enforces an embryonic state, while ablating *Sox11* along with *Sox4* accelerates differentiation. We further show that loss of *Sox11* and *Sox4* impairs cell migration in vitro and wound repair in vivo. Moreover, transcriptomic profiling of *Sox11* induced and *Sox11/Sox4* ablated epidermis shows that close to half of the SOX11- and SOX4-regulated genes are embryonic signature genes, indicating that SOX11 and SOX4 promote the expression of the embryonic program and the repression of the differentiation program. ChIP-seq analysis further illustrates that SOX11 and SOX4 directly regulate genes that govern differentiation and cell motility. Additionally, amongst the embryonic genes that are wound-induced, we identify the actin-bundling protein Fascin1

(FSCN1) as a direct target of SOX11 and SOX4 that facilitates cell migration.

In summary, our study identifies a molecular program employed in embryonic epidermal development that is reactivated during wound repair. We demonstrate that SOX11 and SOX4 play a critical role in this process.

## Results

**Embryonic epidermal genes are upregulated at the wound edge.** To explore the similarity between wound-induced and embryonic epidermal signature genes, we first set out to compile the embryonic epidermal gene signature. Embryonic epidermal cells have the potential to differentiate into cells of the stratified epidermis, hair follicles, and sebaceous glands, whereas postnatal basal cells are more restricted in their lineage fate and only give rise to cells in the stratified epidermis[1]. We performed microarray analysis of mRNAs isolated from two developmental stages using Agilent Sureprint G3 Mouse GE Microarray. We prepared mRNAs from basal epidermal cells from *K14-H2BGFP* transgenic pups at E13.5 and P4, which were isolated based on their positive expression of the cell surface marker α6-integrin and keratin-14-regulated GFP. The compiled heat map from our microarray data showed distinctive clustering of genes that are differentially expressed between the two developmental stages (Fig. 1a). We identified E13.5 epidermal signature genes as genes that are either upregulated or downregulated at E13.5 relative to P4 ≥ 1.5 log2 fold-change and a false discovery rate (FDR) < 0.05.

We next compared E13.5 epidermal signature genes to previously published wound edge epidermal signature genes[4]. The wound-induced epidermal signature genes were generated from comparative RNA-seq analysis of hair follicle stem cells (HFSCs) from unwounded skin and HFSCs and their progeny around the biopsied-wounded area 7 days post wounding. We found that genes upregulated at the wound edge overlapped significantly with genes upregulated at E13.5 ($p = 5.9$e-38, $R = 1.6$). Conversely, genes downregulated at the wound edge and at E13.5 also showed significant overlap ($p = 3.3$e-55, $R = 1.8$) (Fig. 1b). We also used gene set enrichment analysis (GSEA) to show that the embryonic gene signature is enriched in the wounded epidermis (log2 fold-change ≥ 1.5 and FDR < 0.05) (Fig. 1c). These overlapping genes are particularly enriched in gene ontology (GO) categories such as biological processes implicated in cell migration, extracellular matrix organization, as well as embryonic morphogenesis (Fig. 1d).

We were specifically interested in the transcriptional regulation of the genes shared between embryonic and wounded adult epidermis. We identified 32 transcription factors out of the 621 genes upregulated at both E13.5 and the wound edge (Supplementary Table 1). In particular, *Sox11*, out of all the *Sox* family members, shows the highest enriched expression at the embryonic stage (Fig. 1e). We found SOX11 protein uniformly expressed in E13.5 epidermal cells, sparser at later stages, and undetectable by birth (Fig. 1f), consistent with the in situ hybridization data from Allen Brain Atlas (http://developingmouse.brain-map.org/gene/show/20428). We also saw SOX11 protein expression induced at the skin wound edge (Fig. 1g) in agreement with the wound edge gene signature[4].

**Induction of SOX11 represses epidermal differentiation.** Since SOX11 expression in the epidermis is turned off by birth, we sought to investigate the effect of SOX11 induction in postnatal skin. We generated a transgenic line expressing a tet-inducible FLAG-epitope tagged *Sox11* (*TRE-Sox11*, schematic in Fig. 2a) to be used with the established *K14-rtTA* line, which expresses the Tet repressor fused with the transactivator VP16 under the

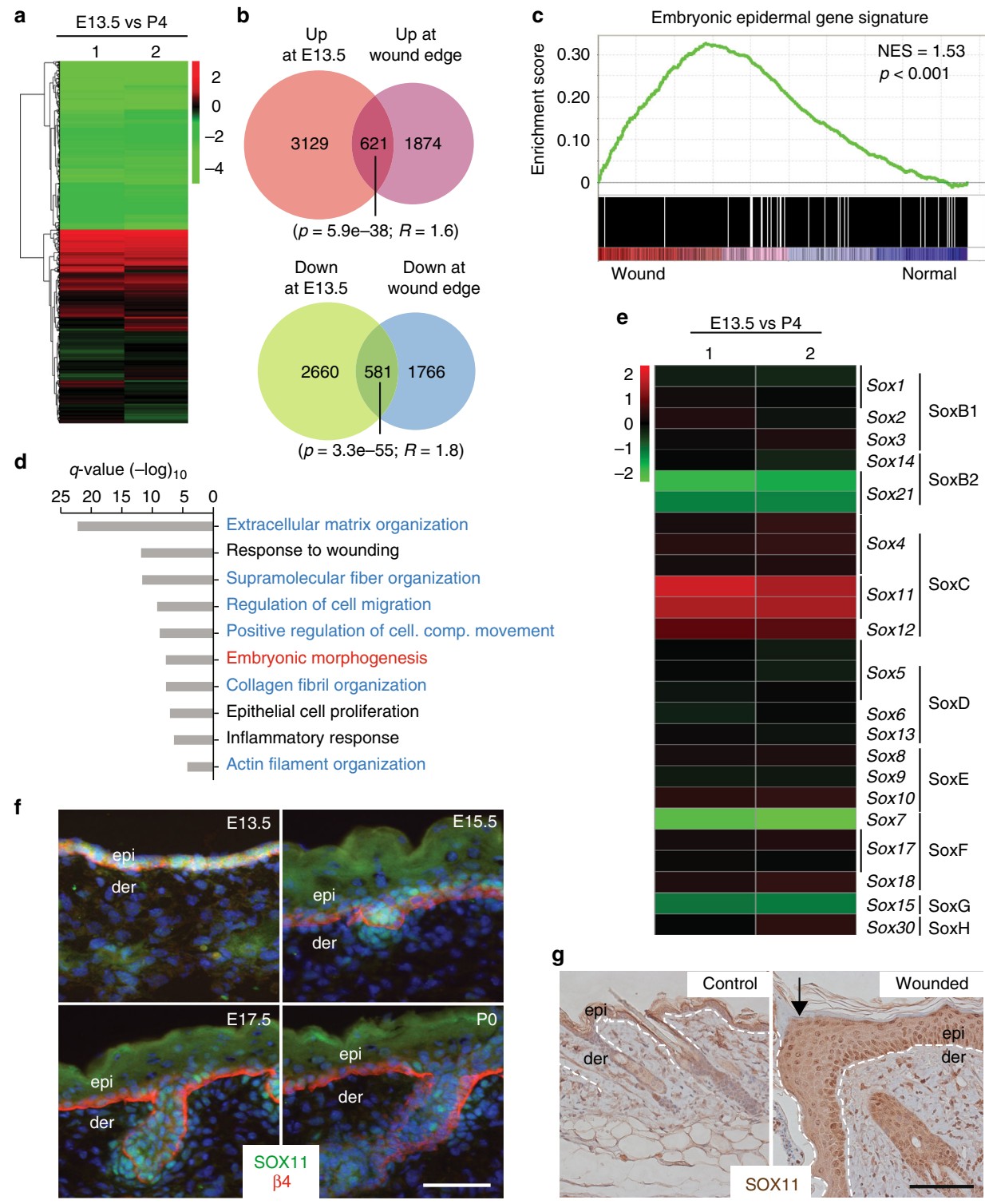

keratin 14 promoter[23]. We confirmed that mice carrying both transgenes express SOX11 in a tightly regulatable manner, as SOX11-FLAG is undetectable without doxycycline (Dox) administration but is expressed in keratin 14 positive cells as early as 6 h post Dox (Fig. 2b).

Using this inducible model, we placed the pregnant mothers carrying double (K14-rtTA;TRE-Sox11) and single (K14-rtTA or TRE-Sox11) transgenic embryos on a Dox diet. By E18.5, the epidermis is fully stratified and has formed a functional skin barrier[24]. Using an X-gal exclusion assay to evaluate the skin barrier function, we found that E18.5 wild-type embryos were impermeable to X-gal as expected while embryos overexpressing SOX11 were permeable to X-gal and consequently turned blue (Fig. 2c).

We next induced expression of SOX11 in newborn pups and found that epidermis expressing SOX11 show reduced levels of intermediate differentiation marker KRT1 (keratin 1) and late differentiation markers LOR (loricrin) and FLG (filaggrin) (Fig. 2d). Moreover, expressing SOX11 in postnatal epidermis reactivates expression of KRT18 (keratin 18), a marker of the

**Fig. 1** Embryonic genes including *Sox11* are induced at the wound edge. **a** Heat map of clustering of differentially expressed (FDR < 0.05 and fold change > 1.5) probesets in epidermal cells at embryonic day 13.5 (E13.5) relative to epidermal basal cells at postnatal day 4 (P4), from two-color microarray data of two biological replicates of each developmental stage. **b** Venn diagrams depicting the overlap of embryonic epidermal signature genes (genes with a log2 fold-change in expression of ≥1.5 and FDR < 0.05 in E13.5 epidermal cells relative to P4 epidermal cell in the basal layer) with wound edge epidermal signature genes[4]. Venn diagram hypergeometric *p* values and the enrichment level (*R*) of the overlaps are indicated. **c** GSEA showing enrichment of embryonic gene signature in wounded epidermis. **d** Top ten GO biological processes (Metascape, *q* values < 0.05) of intersecting genes differentially expressed in both embryonic epidermis at E13.5 and at adult wound edge. Cytoskeletal organization and motility related GO biological processes (blue) as well as embryonic morphogenesis (red) are significantly enriched. **e** Heat map showing relative expression of *Sox* family genes in E13.5 epidermal cells versus P4 basal epidermal cells. **f** Immunofluorescence analysis of SOX11 expression in skins at the indicated developmental stages with β-4 integrin demarking the epidermis from dermis and Hoechst 33342 dye (blue) counterstaining nuclei. **g** Immunohistochemical analysis of SOX11 expression in normal and 5-day post-wounded adult skin. Black arrow points to the wound edges. Epi, epidermis; der, dermis. Scale bars, 50 μm (**f**) 100 μm (**g**). Images in panels **f** and **g** are representative images seen across the skin sections of two biological replicates. Source data for panels **a**–**c** are provided as a Source Data file

single-layer ectodermal cells, as well as other markers expressed in embryonic and not postnatal stratified epidermis including TCF7L1 and TCF7L2[23,25] (Fig. 2e). Since pups expressing SOX11 die after several days of induction, we grafted skins of tet-inducible *Sox11* newborn and control newborn onto nude recipient mice to assess the long-term effect of SOX11. At day 28 post grafting, a full coat of hair was seen on the wild-type grafted skin while little hair was detected on the grafted skin where SOX11 was induced (Fig. 2f). H&E and Oil red O staining of grafted skins show that prolonged expression of SOX11 blocks hair follicle and sebaceous gland development (Fig. 2g–h), in addition to blocking differentiation of the stratified epidermis, as shown by the drastically reduced expression of LOR and FIL (Fig. 2i, j).

As keratinocytes can be induced to differentiate in vitro with high concentrations of calcium[26], we also evaluated the effect of SOX11 expression on the calcium-induced differentiation of primary keratinocytes. We verified that SOX11-FLAG protein is induced in keratinocytes from tet-inducible *Sox11* line when treated with Dox as expected (Supplementary Fig. 1a). Untreated control or tet-inducible *Sox11* keratinocytes differentiate normally in response to calcium, as shown by their flattened morphology and increased expression of differentiation markers. However, when treated with Dox, these Dox-induced SOX11 expressing cells are more resistant to differentiation, reflected by their undifferentiated morphology and their lower expression of differentiation markers and regulators, including *Dsg1a*, *Casp14*, *Tincr*, and *Cdsn* (Supplementary Fig. 1b, c).

**SOX11 induction promotes embryonic signature genes.** Since SOX11 represses the epidermal differentiation while inducing expression of early embryonic epidermal progenitor cell markers, we wished to identify the global gene expression changes caused by SOX11. We isolated stratified epidermis from tet-inducible *Sox11* transgenic pups (*K14-rtTA;TRE-Sox11*) or control pups (*K14-rtTA*) that have been injected with doxycycline for 12 h, and processed mRNAs for transcriptomic analysis. Based on two biological replicates of each genotype, we identified 1299 genes that were altered by log2-fold change > 1.5 by the induction of SOX11, where 854 were upregulated and 445 downregulated (Fig. 3a). The top GO biological processes affected by SOX11 were cytoskeletal/ECM organization, cell migration/adhesion, differentiation, and wound healing (Fig. 3b).

To evaluate whether SOX11 expression controls the embryonic state, we compared genes altered by SOX11 induction to the embryonic signature genes. We found a statistically significant overlap between genes upregulated by SOX11 and genes highly expressed at the early embryonic stage (Fig. 3c). Conversely, genes downregulated by SOX11 also showed significant overlap with genes downregulated at E13.5 (Fig. 3c). We found that 385 genes out of 854 genes (~45%, *p* < 1e-10, *R* = 2.9) upregulated by SOX11 expression were genes upregulated at E13.5. Similarly, 221 genes out of 445 genes (~50%, *p* < 1e-10, *R* = 3.7) downregulated by SOX11 expression were genes downregulated at E13.5. In contrast, much less overlap was observed between genes upregulated by SOX11 and downregulated at E13.5 and *vice versa* (*p* = 0.07, *R* = 1.1 and *p* = 0.84, *R* = 0.9 respectively). Overall, close to half of the genes upregulated by SOX11 belong to embryonic induced signature genes, and an equal percentage of genes downregulated by SOX11 belong to the embryonic repressed signature genes (Fig. 3d). Thus, induction of SOX11 enforces expression of embryonic epidermal signature genes, in line with our observation that SOX11 represses differentiation and induces expression of embryonic epidermal markers (Fig. 2).

**Ablation of *Sox11* and *Sox4* induces premature differentiation.** We next sought to evaluate the consequence of ablating *Sox11* in epidermal development. *Sox11* cKO (*K14cre;Sox11*^fl/fl^) mice were born with no obvious phenotype except for their abnormal eyelid closure also seen in the constitutive KO[19] (Supplementary Fig. 2a). Since SOXC members can compensate for one another[17,21,27], we examined the expression of other SOXC members in epidermal cells. Through quantitative real-time PCR analysis of mRNAs isolated from basal cells of E13.5 and P4 epidermis, we found that while expression of *Sox12* is negligible, both *Sox4* and *Sox11* are highly expressed in E13.5 epidermal cells with *Sox4* at a higher level than *Sox11*. By P4, *Sox4* expression is decreased by 2 fold whereas *Sox11* is undetectable (Fig. 4a).

To uncover if *Sox4* indeed has compensated for the loss of *Sox11* in embryonic skin, we examined mice lacking both *Sox11* and *Sox4* (*K14cre;Sox11*^fl/fl^;*Sox4*^fl/fl^). We found that dcKO pups were born at a Mendelian ratio but died shortly after birth, without detectable milk spots (Supplementary Fig. 2b). Mice lacking *Sox4* appeared very similar to the wild type, but upon closer examination their hair coat seemed shorter (data not shown).

Since SOX11 expression dramatically decreases by E17.5, when the stratum corneum is completely formed, we sought to determine whether the downregulation of SOX11 contributes to differentiation of the stratified epidermis and skin barrier formation. Using the X-gal exclusion assay[28], we compared the level of X-gal penetration in E16.5 embryos that are wild type or lacking *Sox11*, *Sox4*, or both. At E16.5, wild-type embryos have not formed a complete skin barrier as expected and hence turned blue after X-gal incubation. Embryos lacking either *Sox11* or *Sox4* displayed a similar level of X-gal penetration, while those lacking both genes excluded X-gal in most parts of their bodies (Fig. 4b), indicating that deficiency of both genes induces premature epidermal differentiation and thus earlier barrier formation.

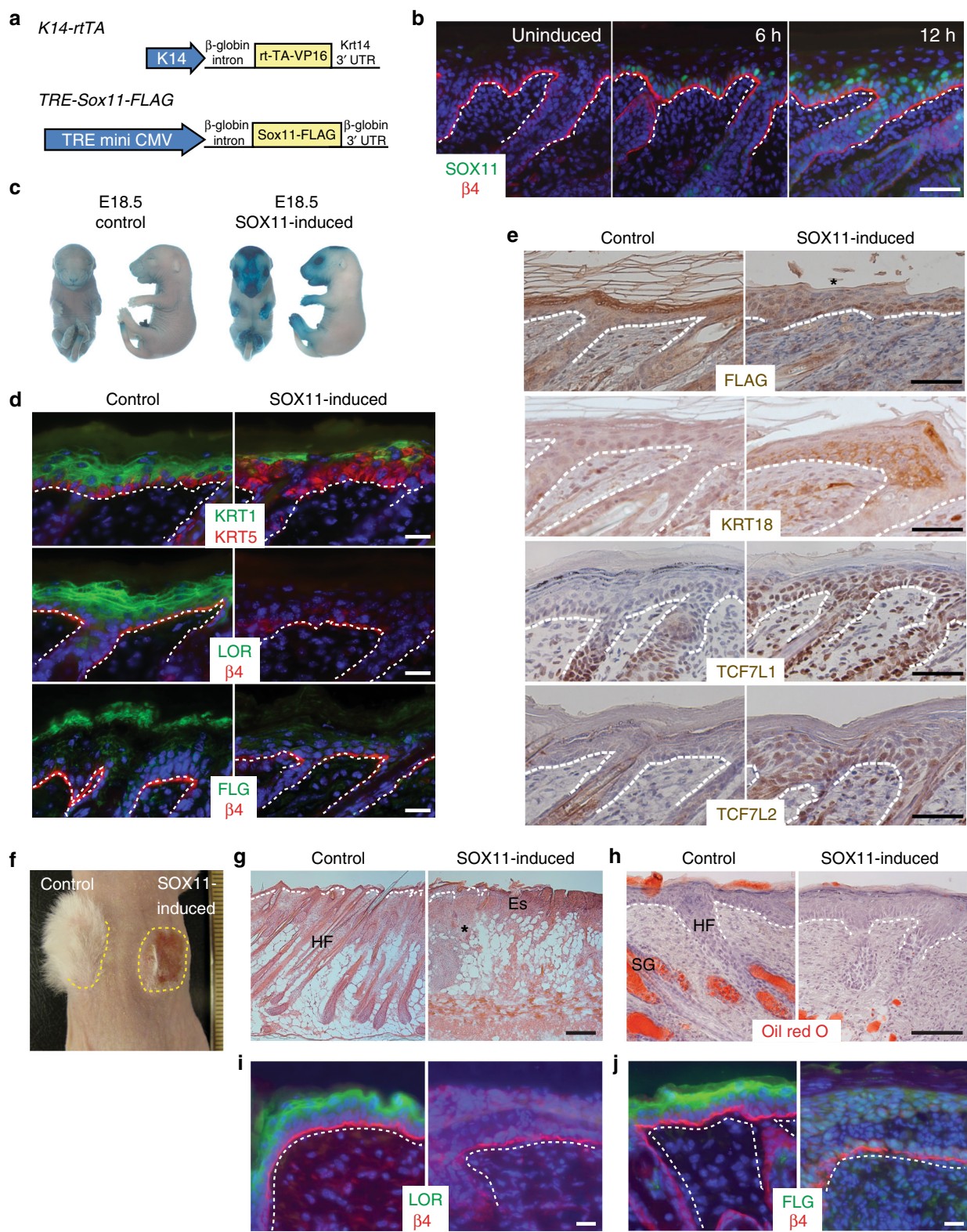

To further validate that deficiency of *Sox11* and *Sox4* accelerates differentiation, we examined the expression of differentiation markers of the stratified epidermis of dcKO embryos. Consistent with the X-gal-exclusion assay results, the level of the late differentiation markers FLG and LOR was higher in the dcKO than in the wild-type or the single knockout embryos (Fig. 4c, Supplementary Fig. 2c), while the level of the early differentiation marker KRT1 showed little change. This

accelerated differentiation was not due to a decrease in proliferation, as dcKO epidermis showed a similar amount of BrdU incorporation (Fig. 4d).

Although *Sox11* is not expressed in postnatal epidermal basal cells, it is induced once cultured in vitro (Fig. 4e), like many wound-induced genes[29]. Interestingly, the differential level in expression of the three SOXC members in vitro recapitulates the differential level seen in vivo at E13.5 (Fig. 4a). We assessed the

**Fig. 2** SOX11 overexpression inhibits epidermal differentiation. **a** Constructs used to generate transgenic mice expressing epithelial-specific *Sox11* under the control of tetracycline and its derivatives. **b** Immunofluorescence analysis of expression of FLAG-tagged SOX11. Four-day old *K14-rtTA;TRE-Sox11-FLAG* mice were injected intraperitoneally with Dox for the indicated time prior to skin isolation. Dashed line demarks the border between the epidermis and dermis. **c** Images of embryos of specified genotypes in X-gal exclusion assay. The pregnant females were on Dox-containing diet until the embryos were collected at E18.5. **d** Immunofluorescence analysis of differentiation markers in postnatal epidermis expressing SOX11. After birth, the mother was on Dox-containing diet for 5 days and skin sections from *K14-rtTA* (control) and *K14-rtTA;TRE-Sox11-FLAG* pups were immunostained with the indicated antibodies. KRT5 (keratin5); KRT1 (keratin 1); LOR (loricrin); FLG (filaggrin). **e** Immunochemical analysis of embryonic makers in control and SOX11-FLAG expressing epidermis with the indicated antibodies. KRT18 (keratin 18); TCF7L1 (transcription factor 7 like 1); TCF7L2 (transcription factor 7 like 2). (**f–j**) Effect of long-term SOX11 overexpression in postnatal skin. Dorsal skins from *K14-rtTA* (control) and *K14-rtTA;TRE-Sox11-FLAG* newborns were grafted in pair onto *Nude* mice, which were put on a diet containing Dox day 10 post grafting. *n* = 2. **f** Image of engrafted mouse 28-day post grafting with engrafted area demarcated by yellow dashed lines. **g** H&E staining of the sectioned pair of skins showing stunted hair follicles (*) in SOX11-induced skin. **f** Oil red O staining marks lipids, enriched in sebaceous glands. Immunofluorescence analysis of differentiation markers LOR (**i**) and FIL (**j**) in grafted skin. White dashed lines mark the border between epidermis and dermis. HF, hair follicles; Es, eschar; SG, sebaceous glands. Scale bars, 50 µm (**b**, **e**) 20 µm (**d**, **i–j**), 100 µm (**g**, **h**). All images reflect the same result seen across the entire embedded skin section from 2 (**b**, **c** and **f–j**) or 3 (**d** and **e**) biological replicates

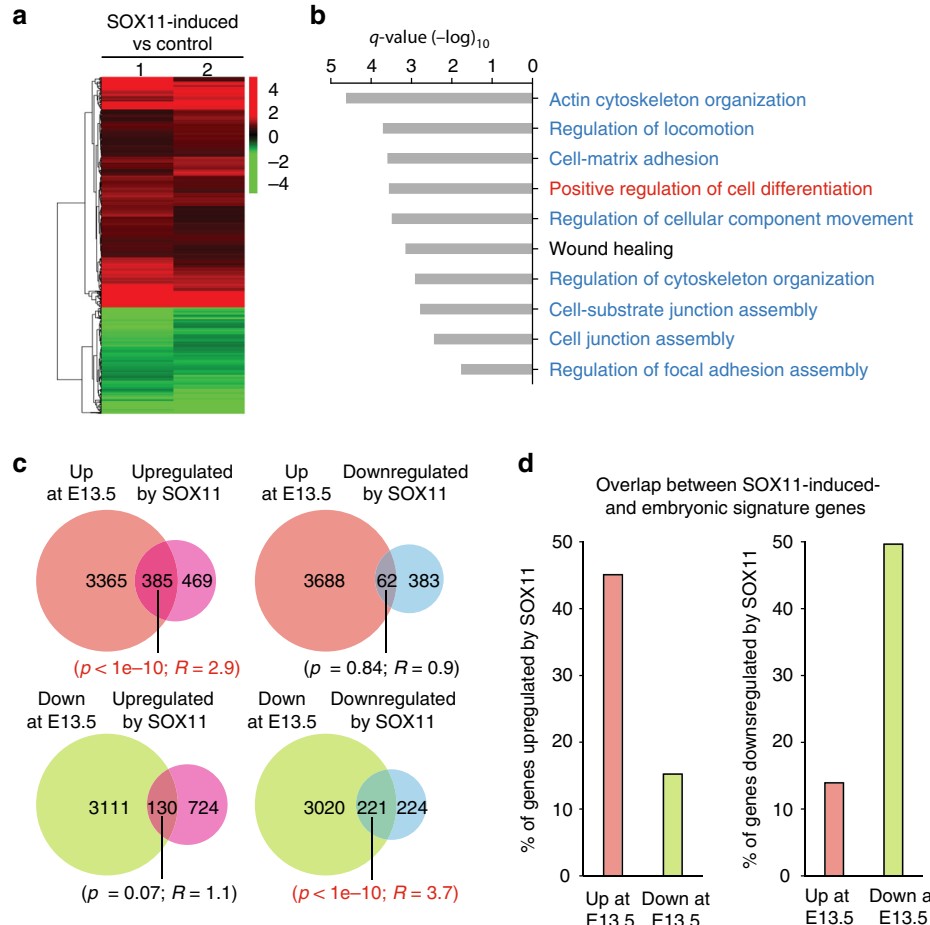

**Fig. 3** SOX11 overexpression induces embryonic transcriptional program. **a** Heat map from two-color microarray analysis depicts clustering of differentially expressed (FDR < 0.05 and fold change > 1.5) probesets in *K14-rtTA;TRE-Sox11* epidermis relative to control *K14-rtTA* epidermis at P4 after 12 h of Dox induction. Data from two biological replicates of each genotype. **b** Top ten GO biological processes of the differentially expressed genes in SOX11-induced epidermis. Cell organization and movement related (blue) and epidermal differentiation (red) processes are among the most significantly enriched. **c** High overlap of SOX11-induced signature genes and embryonic epidermal signature genes. The Venn diagrams show overlapping between E13.5 epidermal signature genes and genes changed by SOX11-induction (genes with log2-fold change > 1.5 in both microarrays). Venn diagram hypergeometric *p* values and the enrichment level (*R*) of the overlaps are indicated below each Venn diagram, with statistically significant values highlighted in red. **d** Percentage of genes altered by SOX11 overexpression overlapping with embryonic signature genes. Source data for panels **a**, **c** and **d** are provided as a Source Data file

effect of ablating *Sox11*, *Sox4*, or both on the response to calcium, and found that dcKO cells express a higher basal and calcium-induced level of certain differentiation markers and regulators (*Dsg1a, Casp14, Tincr, and Cdsn*) than the control cells (Fig. 4f).

It should be noted that the in vitro data do not completely recapitulate the in vivo observation, as certain differentiation markers that were upregulated in dcKO embryonic epidermis such as *Inv*, *Lor*, and *Lce1a* were not upregulated in vitro. Nevertheless, our in vitro and in vivo data together strongly suggest that deficiency of both *Sox11* and *Sox4* affects expression of differentiation genes, which likely predispose cells to differentiate more readily.

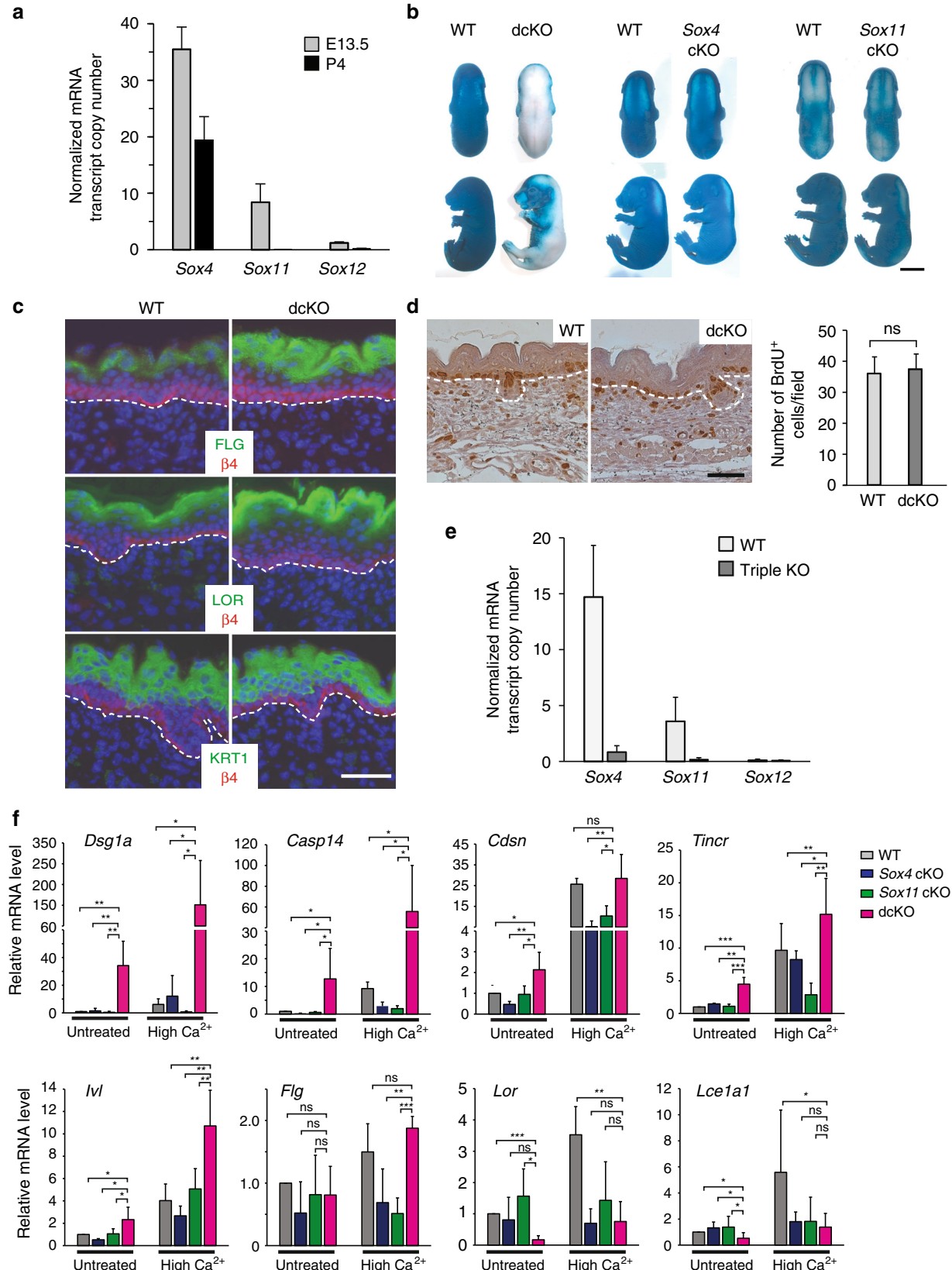

**SOX11 and SOX4 regulate embryonic epidermal signature genes**. We next set out to explore the molecular mechanism underlying SOX11 in epidermal differentiation. Since ablating both *Sox4* and *Sox11* induces accelerated epidermal differentiation by E16.5, we assessed the change in gene expression through transcriptionally profiling E16.5 epidermis lacking *Sox11*, *Sox4*, or both. Using epidermis isolated through dispase digestion, we first conducted RT-qPCR expression analysis for *Sox4*, *Sox11, Flg*, *Krt1*, and *Krt5* to verify (1) loss of *Sox4* and/or *Sox11* from the respective knockouts, (2) increase in *Flg* in dcKO, and (3) similar

**Fig. 4** Ablation of *Sox11* and *Sox4* induces premature epidermal differentiation (**a**) mRNA transcript copy number of SOXC class genes from FACS-purified E13.5 epidermal and P4 basal epidermal cells normalized to the housekeeping gene *Mrpl19*. Data are the mean ± SD. *n* = 2 biological replicates. **b** Images of E16.5 embryos of indicated genotypes in X-gal exclusion assay. Independent biological replicates of each genotype pair *n* = 3 *Sox4* cKO-control pairs, 5 *Sox11* cKO-control pairs, and 4 dcKO-control pairs in independent experiments. **c** Immunofluorescence analysis of epidermal differentiation marker expression in WT and dcKO at E16.5 (representative images from *n* = 3 biological replicates). FLG (filaggrin) and LOR (loricrin) are late differentiation markers expressed in termi80 (WT), 63 (*Sox4* cKO), 83 (*Sox11* cKO), or 88 (dcKO).nally differentiated keratinocytes in the epidermal outer layer; KRT1 (keratin 1) is an early differentiation maker expressed in cells of the intermediate epidermal layers. **d** BrdU incorporation analysis of WT and dcKO embryos. Pregnant mice were administered with BrdU for 4 h prior to embryo isolation. Skin sections of embryos at E16.5 were immunostained with anti-BrdU antibody. Representative sections are shown (left panel) and BrdU + cells in 13 fields were quantified (right panel). Data are the mean ± SD. *n* = 2 independent experiments. *p* = 0.51. **e** mRNA transcript copy numbers of SOXC class genes expressed in primary keratinocytes, normalized to the housekeeping gene *Mrpl19*. Cells were isolated from neonatal skins of WT and *Sox4/11/12*-triple knockout littermates. Data are the mean ± SD. *n* = 2 triple KO and 6 wild-type (WT) biological replicates. **f** qRT-PCR analysis of epidermal differentiation gene expression in WT, *Sox11* cKO, *Sox4* cKO, or dcKO primary keratinocytes. Keratinocytes were cultured in regular media or with the addition of calcium at 1.5 mM for 24 h. Data are the mean ± SD. *n* = 5 biological replicates. *$p < 0.05$, **$p < 0.01$, ***$p < 0.001$; ns, not significant (Student's one-tailed *t*-test). Scale bars, 5 mm (**a**), 50 μm (**c**, **d**). Source data for panels **a** and **d**–**f** are provided as a Source Data file

levels of *Krt1* and *Krt5* between knockouts and wild-type controls (Fig. 5a). We then carried out the two-colored microarray analysis using these validated mRNAs from two biological replicates of each genotype.

Among the probesets that show an increase or decrease in expression at log2-fold change > 1.5 or greater with FDR <0.05 over the wild-type control, we found that over 60% of the probesets show altered expression exclusively in dcKO whereas only approximately 15% show changed expression in *Sox4* cKO or *Sox11* cKO alone (Fig. 5b). Since ablation of both *Sox11* and *Sox4* altered expression of many more genes than the ablation of either gene alone, our result suggests that *Sox11* and *Sox4* have significant compensatory roles in epidermal cells.

We identified 1065 probesets specifically altered only in the dcKO (Fig. 5c). Since deficiency of both *Sox11* and *Sox4*, but not either alone, accelerated differentiation in embryonic epidermis, we focused only on the probesets that were altered by the ablation of both genes and not by either alone. In our GO enrichment analyses of these genes, we found two major groups of biological processes significantly affected by the loss of both *Sox11* and *Sox4*. The first group is linked to skin development, which includes keratinization and sphingolipid metabolism, and the second group is involved with functions related to cytoskeletal, extracellular matrix and supramolecular fiber organization (Fig. 5d).

Since *Sox11* is especially enriched at embryonic epidermal cells and its ablation along with *Sox4* accelerates differentiation, we investigated whether ablating *Sox11* and *Sox4* shifts the developmental stage of the cells. We compared genes with altered expression uniquely in dcKO epidermal cells to genes altered between E13.5 and P4 developmental stages (Fig. 5e). We found that 63% of the genes decreased in dcKO are those highly expressed at E13.5 ($p < 1e-10$, enrichment level $R = 4.3$) while 55% of the genes increased in dcKO are those downregulated at E13.5 ($p < 1e-10$, $R = 4.4$) (Fig. 5f). In contrast, the overlaps between downregulated genes in dcKO and in E13.5 ($p = 0.91$, $R = 0.8$) or between upregulated genes in dcKO and in E13.5 ($p = 1.0$, $R = 0.6$) are not significant. These data suggest that SOX4 and SOX11 induce the expression of genes found in embryonic progenitors and repress genes found in the later developmental stage.

We next sought to identify the direct target genes of SOX11 and SOX4 using chromatin immunoprecipitation and high throughput sequencing (ChIP-seq). As of now, very few direct targets of SOX4 and SOX11 have been validated, other than *Tubb3* and *Tead2*, as well as *Dcx* and *Prox1* in neurons[17,30,31]. Since we could not identify antibodies against SOX4 or SOX11 that work effectively for ChIP, we took advantage of a well-established antibody against the FLAG-epitope to immunoprecipitate SOX11-FLAG or SOX4-FLAG. To avoid ChIP-seq signal loss due to competition from endogenous SOX4 and SOX11 in keratinocytes, we transduced double null keratinocytes with lentiviral vector expressing tet-inducible FLAG-epitope tagged *Sox11* or *Sox4*. We treated the stably transduced cells with Dox for 24 h prior to harvesting cells for the ChIP experiment. After verifying that both SOX11-FLAG and SOX4-FLAG were induced (Fig. 6a, Supplementary Fig. 6a) and bound to known binding sites in the *Tead2* and *Tubb3* genes as expected (Fig. 6b), we used validated ChIP samples of two biological replicates for library preparation and sequencing. We identified a drastically higher number of SOX11-peaks (21270) than SOX4-peaks (2669) (Fig. 6c), although both transcription factors have similar enrichment (Fig. 6d). We suspect that the far fewer SOX4-bound sites could be due to SOX4 having fewer available binding sites genome-wide, a lower affinity for the binding sites, and/or shorter occupancy at binding sites.

Overall, we found that SOX4 and SOX11 bind many of the same genomic regions, with 94% of SOX4-bound regions overlapped with SOX11-bound regions (Fig. 6c). In addition, both proteins display similar genome-wide binding characteristics across promoters and putative enhancers (Fig. 6e). Through motif enrichment analysis, we detected a high degree of similarity in SOX4 and SOX11 binding motifs. Interestingly, regions containing binding sites to SOX11 and to a lesser extent to SOX4, are highly enriched for the AP-1 motif (Fig. 6f). Analysis of the overlapping SOX4 and SOX11 peaks indeed shows a strong enrichment for SOX4-like motifs flanked by AP-1, with little apparent TEAD motif co-enrichment (Fig. 6g, h).

We sought to validate our ChIP-seq data by evaluating the effect of SOX11 and SOX4 on transcription of a few target genes identified by ChIP-seq using a luciferase reporter assay. We cloned out genomic regions containing the SOX11/4 ChIP-seq peaks of *Tead2*, *Fscn1*, *Fblim1*, *Marcksl1*, and *Pxdn* and placed them in either forward or reversed sequence orientation upstream of a luciferase reporter gene. Of the five selected targets, SOX11 activates transcription with all five selected enhancer regions placed in either direction (Fig. 6i), while SOX4 activates transcription with only one of the five. The result is not entirely surprising given that SOX4 has been reported to show lower transactivating activity in vitro assays[16].

We also sought to further validate our ChIP-seq data from SOX11-induced keratinocytes by performing ChIP-qPCR experiment on epidermal cells freshly isolated from 4-day old transgenic pups that have been induced to express SOX11-FLAG for 24 h. The fixed freshly isolated cells required much harsher sonication condition than the one used in our ChIP-seq experiment with keratinocytes to successfully generate 200–900

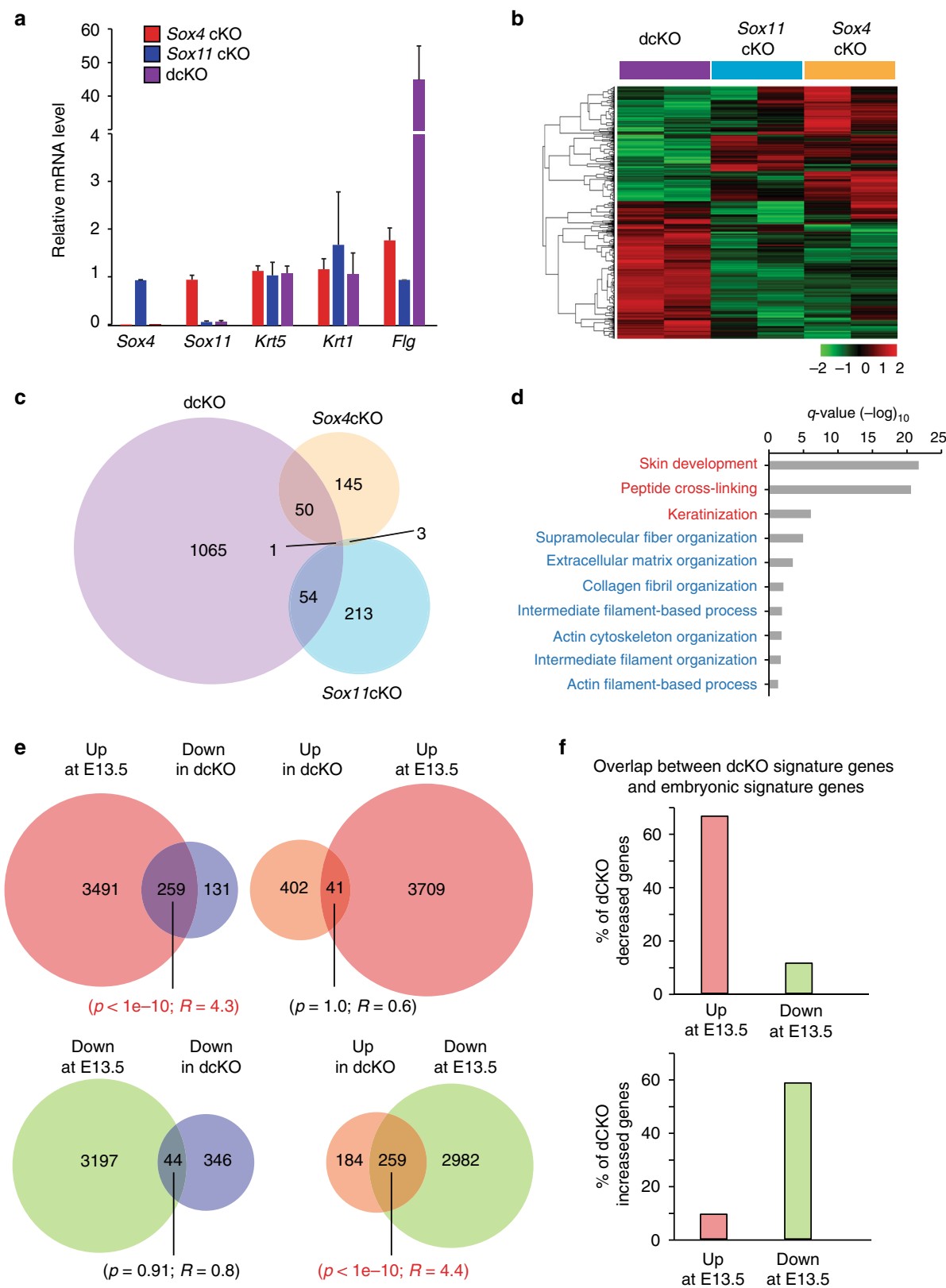

bp fragments required for optimal ChIP-qPCR. Perhaps due to the harsher sonication condition, which reduces the level of FLAG-epitope SOX1 protein in the lysate (data not shown), results from our ChIP-qPCR analysis of freshly isolated cells did not completely recapitulate our ChIP-seq data. We detected enrichment mostly at regions that show high SOX11 ChIP-seq

peaks (higher than 15) such as those found in *Tes, Lfng, S100a10, Smad3, Tead2*, and not at sites with smaller peaks such as those found in *Lvl, Sprrd2, Flg2, Fscn1, Lgals1* (Supplementary Fig. 3). Overall, most selected targets with high ChIP-seq peaks were validated by ChIP-qPCR analysis of freshly isolated cells from SOX11-induced pups, with a few exceptions such as one region

**Fig. 5** Identifying downstream effectors of SOX11 and SOX4 in embryonic epidermis. **a** qRT-PCR analysis of isolated epidermis from *Sox11* cKO, *Sox4* cKO, or dcKO and their WT littermate embryos at E16.5 with two biological replicates per genotype. The values are gene expression levels in the knockouts relative to their respective wild-type controls. Data are the mean ± SD. $n = 2$ biological replicates. **b** Two-color microarray analysis was performed on E16.5 epidermis lacking either *Sox11* or *Sox4*, or both genes, as well as on wild-type littermate, which is used as a baseline control. Heat map shows clustering of probesets with significant differential expression (FDR < 0.05 and fold change > 1.5) in dcKO, *Sox11* cKO or *Sox4* cKO epidermis relative to their WT littermate controls. $n = 2$ biological replicates. **c** Venn diagram showing the overlap of probesets significantly changed in dcKO, *Sox11* cKO or *Sox4* cKO epidermis. **d** Cell organization/motility (blue) and epidermal development (red) are among the top GO biological processes found in the differentially expressed genes in dcKO epidermis at E16.5 (as determined by Metascape, q values < 0.05). **e** Venn diagrams (Top panel) show the overlaps of genes up- or downregulated (log2-fold change ≥ 1.5 and FDR < 0.05) in E16.5 dcKO epidermis with genes up- or downregulated in E13.5 epidermal cells as compared to P4 basal epidermal cells. Hypergeometric p values and the enrichment level (R) of the overlap show statistical significance between specified groups (highlighted in red). **f** Graph shows the percentage of genes altered in dcKO overlapping with embryonic signature genes. Source data are provided as a Source Data file

with small SOX11 ChIP-seq peak in *Fblim1* that was enriched by ChIP-qPCR or one region in *Nf2l2* with a high SOX11 ChIP-seq peak but not enriched by ChIP-qPCR.

To identify the direct targets of SOX11 and/or SOX4 that contribute to the phenotypes of dcKO, we cross-analyzed the E16.5 dcKO transcriptome with the ChIP-seq data. We selected genes affected by the ablation of both *Sox11* and *Sox4* that are bound to SOX11 and/or SOX4. We opted using binding to SOX11 and/or SOX4 instead of binding to both as a criterion, since SOX4-FLAG shows ~4x lower affinity to positive control sites in known target genes *Tead2* and *Tubb3* (Fig. 6b, d), which suggests that SOX4-ChIP-seq data could contain a much higher number of false negative. This cross-comparison analysis identified 487 genes with altered expression in dcKO that are bound to SOX11 and/or SOX4 (Fig. 7a). By comparing these 487 targets to the embryonic signature genes, we found that 166 of 238 (or 70%) of the bound targets that are downregulated in dcKO are highly expressed in embryonic epidermal cells, and 151 of 249 (or 60%) of the bound targets upregulated in dcKO are lowly expressed in embryonic cells. These genes fall into main GO categories related to cytoskeletal organization, cell motility, and epidermal development (Fig. 7b).

Many of the epidermal development related genes increased in dcKO epidermis encode components of the cornified envelope, such as *Filaggrin* (*Flg*) and late cornified envelope (*Lce*) proteins, which form the skin barrier. These genes are located in the epidermal differentiation complex (EDC), a 2 Mb locus on mouse chromosome 3[32,33]. Our ChIP-seq data demonstrate that both SOX11 and SOX4 bind to multiple sites in this region (Fig. 7c). The loss of function of both genes in the epidermis at E16.5 results in dramatic upregulation of most EDC genes (Fig. 7d, Supplementary Table 2), likely accounting for the accelerated barrier formation (Fig. 4b).

**Ablation of *Sox11* and *Sox4* impairs epidermal cell migration.** Since SOX11 expression is induced upon wounding (Fig. 1f) and loss of function of *Sox11* and *Sox4* affects expression of a significant number of motility related genes (Figs. 5d, 7b), we evaluated the role of SOX11 and SOX4 in epidermal cell migration and wound repair. Using a well established in vitro scratch assay, we found keratinocytes lacking either *Sox11* or *Sox4* migrated at a similar rate to that of wild-type keratinocytes, whereas cells deficient of both genes migrated at a significantly reduced rate (Fig. 8a, b). Moreover, introduction of either *Sox11* or *Sox4* into these double null cells improved their migratory capacity (Fig. 8c), demonstrating that SOX11 and SOX4 play a vital role in keratinocyte migration.

We next employed the splinted wound-healing model, which minimizes the effect of contraction, in order to focus on the in vivo cell migration/re-epithelialization process of wound healing[34]. Since double null pups die neonatally, we grafted newborn skin from dcKO, single cKO, and wild type littermates onto immunodeficient mice, and created a $4 \times 10$ mm splinted full-thickness wound on the grafted skin 8 weeks post grafting (Fig. 8d). By examining microscopic images of the H&E stained skin sections isolated from the wound area 5 days post wounding, we found significantly less re-epithelialization in dcKO skin and little change in the single cKO (Fig. 8e, f).

To identify SOX11- and SOX4-regulated genes responsible for defective cell migration in dcKO keratinocytes, we conducted transcriptomic analysis on primary keratinocytes lacking *Sox11*, *Sox4*, or both (Supplementary Fig. 4a). Ablation of both *Sox4* and *Sox11* altered the expression of many genes not affected by the ablation of either gene alone (Supplementary Fig. 4b). Since only dcKO keratinocytes show impaired migration, we focused on the genes solely changed in dcKO. As many embryonically expressed genes reactivated in response to wounding are particularly enriched in GO categories related to cytoskeletal/ECM organization and cell migration (Fig. 1d), we overlaid the genes in these categories differentially expressed in embryonic and wounded epidermis and found a statistically significant overlap (Supplementary Fig. 4c). We found that 267 out of 895 genes in this group upregulated at the wound edge were genes expressed at higher level at E13.5 versus P4 ($p = 2.3\text{e-}10$, $R > 1$). Conversely, 228 out of 886 genes in this group downregulated at the wound edge were downregulated at E13.5 ($p < 1\text{e-}10$, $R > 1$), while no significant overlap was seen between the motility related genes upregulated at the wound edge and genes downregulated at E13.5 and vice versa ($p > 0.05$, $R \leq 1$ for both). This reveals that a significant number of embryonic cytoskeletal components are redeployed in adult epidermal cells at the wound edge.

Since SOX11 and SOX4 regulate genes involved in cytoskeletal/ECM organization and cell migration in embryonic epidermis (Figs. 5d, 7b), we wished to identify embryonic signature genes regulated by SOX11 and SOX4 that are induced by wounding and contribute to cell migration. By overlaying the genes that were differentially expressed in dcKO keratinocytes with the wound edge epidermal signature genes and E13.5 epidermal signature genes (Fig. 8g), we saw 126 out of the 621 embryonic genes induced at the wound edge that were positively regulated by SOX11 and SOX4 (downregulated in dcKO keratinocytes), and 79 genes downregulated at E13.5 and in the wounded epidermis were negatively regulated by SOX11 and SOX4 (upregulated in dcKO keratinocytes).

To specify how many of these 126 and 79 genes are directly regulated by SOX11 and SOX4 during development as well, we further compared these genes to genes differentially expressed in dcKO E16.5 epidermis and bound to SOX11 and/or SOX4. From this stringent analysis, we identified 25 genes positively regulated and four negatively regulated genes by SOX11 and SOX4 that are directly bound to SOX11 and/or SOX4. Close to half of the genes positively regulated are implicated in cytoskeletal organization and cell migration (Table 1, Supplementary Table 3). Among the

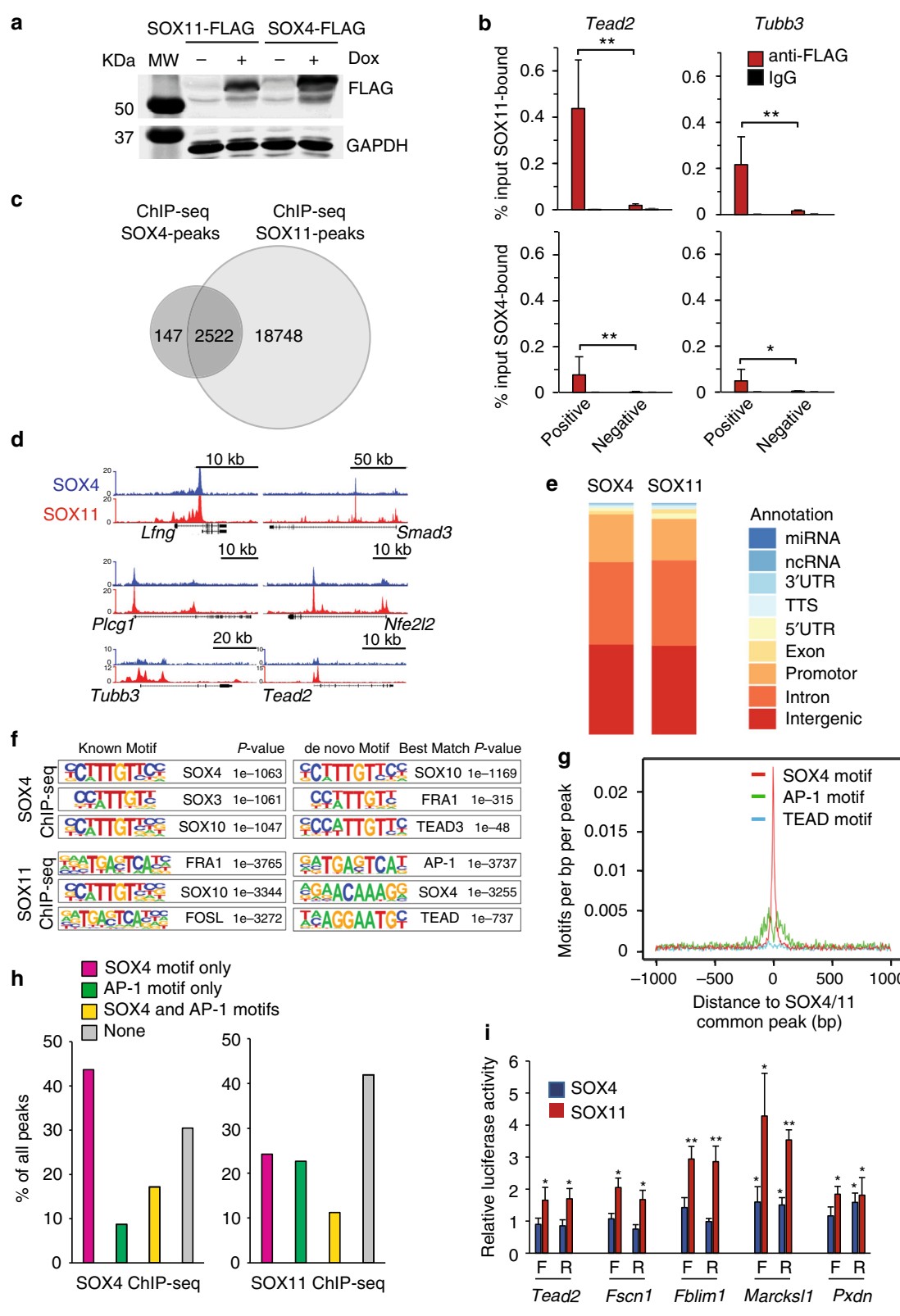

positively regulated genes that are direct targets of SOX11 and/or SOX4, *Fscn1*, *Fblim1*, *Lgals1*, and *Tmsb10* have been documented to be induced in the migrating keratinocytes at the leading edge of wounded murine tail epidermis[35].

**SOX11 and SOX4 induce cell migration partially through *Fscn1*.** Amongst the direct target genes of SOX11 and/or SOX4,

we chose to focus on *Fblim1* and *Fscn1*, which are implicated in cytoskeletal/ECM re-organization. FBLIM1, also known as migfilin, is an adaptor protein localized at focal adhesion sites that is implicated in actin cytoskeleton organization and cell adhesion[36]. *Fascin1* encodes an F-actin bundler that assembles F-actin into parallel bundles, facilitating the formation of actin-based cellular protrusions and cell migration[37].

**Fig. 6** Identification of direct targets of SOX11 and SOX4 by ChIP-seq profiling. **a** Western blot analysis of induced expression of SOX11-FLAG and SOX4-FLAG in keratinocytes. Western blotting was done on isolated GFP+ cells cultured for 24 h with or without Dox. The representative blots shown are from three independent experiments. **b** ChIP-qPCR verifies binding of genomic region containing known SOX4/11 binding sites in the *Tead2* and *Tubb3* genes to FLAG-tagged SOX11 or SOX4 protein. Primers were designed to amplify the known binding sites (positive) or neighboring regions without potential SOX consensus elements (negative). Data are the mean ± SD. $n = 4$ biological replicates. *$p < 0.05$ (Student's one-tailed *t*-test). **c** Overlay of SOX11 and SOX4 ChIP-seq peaks identified from ChIP-seq experiment where lysates from two biological replicates of keratinocytes deficient of both *Sox11* and *Sox4* induced to express SOX11-FLAG or SOX4-FLAG for 24 h and the amount of specified genomic regions bound by SOX11-FLAG or SOX4-FLAG were quantified. **d** Representative ChIP-seq profiles of indicated genes with genomic region binding to SOX11 and SOX4. **e** Annotation of SOX11 and SOX4 genome-wide binding sites. **f** Homer motif analysis for SOX11 and SOX4 ChIP-seq. (Left) Matches to known motifs; and (Right) de novo motif enrichment with best motif matches indicated. FRA1 and FOSL2 are FOS-related members of AP-1 transcription factor complex. **g** SOX4, AP-1, and TEAD motif enrichment around SOX11 and SOX4 binding sites. **h** Ratios of SOX11 or SOX4 ChIP-seq peaks with or without SOX4 and/or AP-1 consensus binding motifs. **i** Genomic regions containing SOX4 and SOX11 ChIP-seq peak from *Tead2, Fscn1, Fblim1, Pxdn,* and *Marcksl1* were cloned and placed in either forward (F) or reversed (R) orientation upstream of the luciferase reporter gene. Luciferase activity was measured and *Firefly* luciferase activity *was* normalized over *Renilla* luciferase activity. Graph shows normalized luciferase activity relative to vector control. $n = 5$ indep*e*ndent experiments. Data are mean ± SD. *$p < 0.05$, **$p < 0.01$ (Student's one-tailed *t*-test). Source data for panels **a**, **b**, **h**, and **j** are provided as a Source Data file

We found FBLIM1 highly localized at focal adhesions in wild type primary keratinocytes but undetectable in dcKO cells (Supplementary Fig. 5a). Consistent with the microarray result, real-time PCR analysis confirmed that deficiency of both *Sox11* and *Sox4*, but not either of the genes alone, causes a dramatic reduction of *Fblim1* expression level in embryonic epidermis and primary keratinocytes (Supplementary Fig. 5b). Although we could not validate its expression in skin due to the lack of commercially available FBLIM1 antibodies that function in skin tissue, we proceeded to use CRISPR/Cas9 to ablate *Fblim1* in primary keratinocytes to determine its role in migration (Supplementary Fig. 5c, 6b). However, we found that the ablation of *Fblim1* has no impact on cell migration (Supplementary Fig. 5d).

We next examined the other potential candidate, *Fscn1*, in skin. We saw FSCN1 abundantly expressed in the embryonic epidermis, drastically reduced at E17.5 and undetectable by birth (Fig. 9a), mirroring the expression of SOX11. FSCN1 expression was drastically reduced in embryonic epidermis lacking both *Sox11* and *Sox4*, but unchanged in single cKO epidermis (Fig. 9b, Supplementary Fig. 5e). Conversely, its expression was induced in SOX11 overexpressing skin (Fig. 9c). Real-time PCR analysis of E16.5 epidermis and cultured keratinocytes further confirmed that expression of *Fscn1* mRNA level is dependent on both SOX11 and SOX4 (Fig. 9d). Consistent with the mRNA expression level, protein level of FSCN1 was drastically reduced in dcKO keratinocytes, but unchanged in single cKO cells (Fig. 9e).

Moreover, like SOX11, FSCN1 is undetectable in postnatal skin but is strongly induced at the wound leading edge (Fig. 9f). Furthermore, FSCN1 was not induced at the wound edge in skin deficient of both *Sox4* and *Sox11* (Fig. 9f, Supplementary Fig. 5f), affirming that FSCN1 expression is under the control of both SOX11 and SOX4. Since FSCN1 functions as a key specific actin cross-linker, which is important for filopodial protrusion and cell migration[37,38], we immunostained WT and dcKO cells with antibodies against VCL (vinculin) and phalloidin to visualize their filopodia. We found dcKO cells contain shortened filopodia, as confirmed by quantification (Fig. 9g, h).

To determine whether the abrogation of *Fscn1* is a contributing factor to the reduced migration rate of dcKO cells, we examined migration of cells whose *Fscn1* is ablated by CRISPR/Cas9 editing (Fig. 9i, Supplementary Fig. 6c). We found that *Fscn1*-ablated cells migrated less than the wild type cells albeit better than cells lacking both *Sox11* and *Sox4* (Fig. 9j). We also saw that ectopic induction of *Fscn1* improved the migration of dcKO cells but did not completely rescue their defect (Fig. 9k, l). These results strongly indicate that *Fscn1* is a downstream effector of SOX11

and SOX4 required for effective cell migration, and also suggest that SOX11 and SOX4 affect cell migration through the regulation of additional direct target genes, which remain to be investigated.

## Discussion

In summary, our discovery that adult epidermal cells at the wound edge are converted to express an embryonic gene program suggests that the induction of an embryonic program is critical for wound repair. We show that the wound-induced reactivation of an embryonic program is driven by SOX11 and SOX4 and that the re-epithelialization step of wound repair requires both SOX11 and SOX4.

While the role of SOX11 in skin was previously unknown, the role of SOX4 in skin has been studied with a *Sox4* knock-in line, where the original *Sox4* locus was replaced with a KI construct flanked with *loxP* sites that contains the *Sox4* cDNA with an IRES-GFP-luciferase[22]. However, these KI mice show a 10-fold decrease in *Sox4* mRNA expression in all tissues and have reduced body weight and size even in the absence of any cre-recombinase. Therefore, the observed phenotype of delayed hair regrowth and wound healing, and high resistance to tumor formation could not be definitively attributed to epidermal deficiency of *Sox4*, leaving the role of *Sox4* in skin unresolved.

Reactivating SOX11 expression enforces an embryonic state and induces embryonic signature genes and loss of both *Sox11* and *Sox4* and not of either gene alone leads to premature differentiation and impaired migration and re-epithelialization. Consistent with their implicated role in repressing differentiation, SOX11 and SOX4 significantly alters expression of embryonic signature genes and downregulate target genes that encode epidermal differentiation regulators and cornified envelope precursors in the EDC cluster. Deficiency of both *Sox11* and *Sox4* but not either gene alone upregulates expression of most of EDC genes (32 out of 43), which likely accounts for the accelerated epidermal differentiation observed in E16.5 dcKO embryos. The genes in the EDC cluster are under control of multiple activators and repressors[39–42]. Repressive factors block the activity of this locus in embryonic progenitors and basal layer cells while the activators regulate concomitant EDC gene expression during skin barrier formation. Unlike other master transcriptional activators of terminal differentiation that are expressed suprabasally, AP-1 is expressed in progenitor and basal cells where its function is impeded by the Polycomb repressor complex containing EZH2[40,43]. AP-1 can regulate the dynamic chromatin architecture of EDC by modulating the conserved human 923-centric EDC chromatin domain (or enhancer)[44], which can physically interact

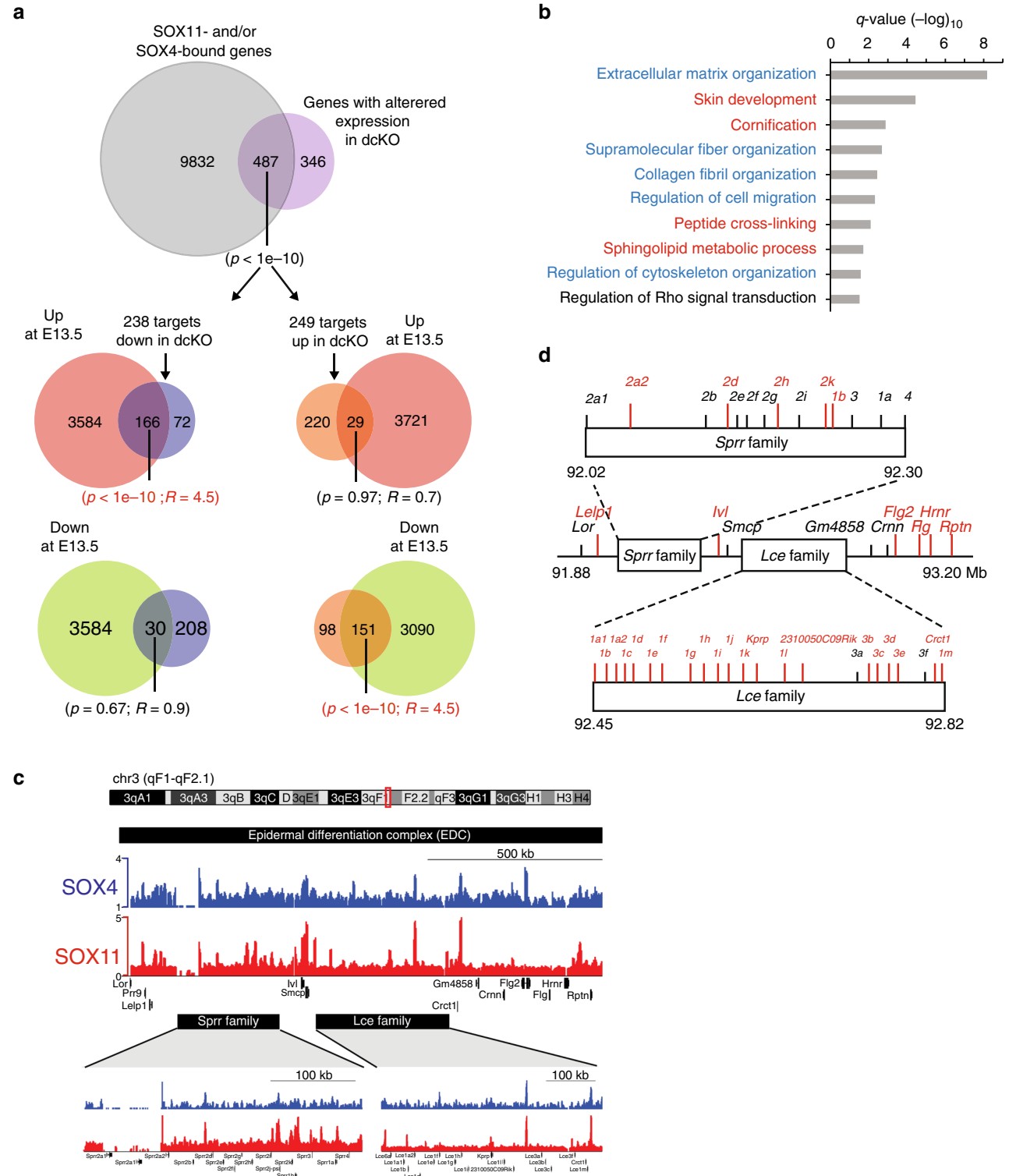

**Fig. 7** Identification of direct targets of SOX11 and SOX4 in embryonic epidermis. **a** Top Venn diagram shows the overlap of SOX4/11-bound targets (ChIP-seq) and transcripts in dcKO E16.5 epidermis whose expression is changed >1.5-fold (log2-fold change) and FDR < 0.05. The bottom Venn diagrams show the overlap between the SOX11- and SOX4- directly regulated targets and the differentially expressed genes in E13.5 epidermal cells. Statistically significant *p* values and the enrichment level *R* values are highlighted in red. **b** GO enrichment analyses of the 487 direct targets of SOX11 and/or SOX4 in embryonic epidermal progenitors (from Fig. 7a), highlighting categories related to cell cytoskeleton and movement (blue) and epidermal development (red), as determined by Metascape, *q* values < 0.05. **c** SOX11 and SOX4 ChIP-seq tracks displaying shared SOX11- and SOX4-bound genomic regions in the EDC locus. *Sprr* and *Lce* family gene clusters are shown at higher magnification below. **d** Schematic representation of the epidermal differentiation cluster (EDC) on mouse chromosome 3. Late-stage differentiation EDC genes that were highly upregulated in dcKO epidermis at E16.5 are marked in red. They are also listed in Supplementary Table 2. Source data for panel **a** are provided as a Source Data file

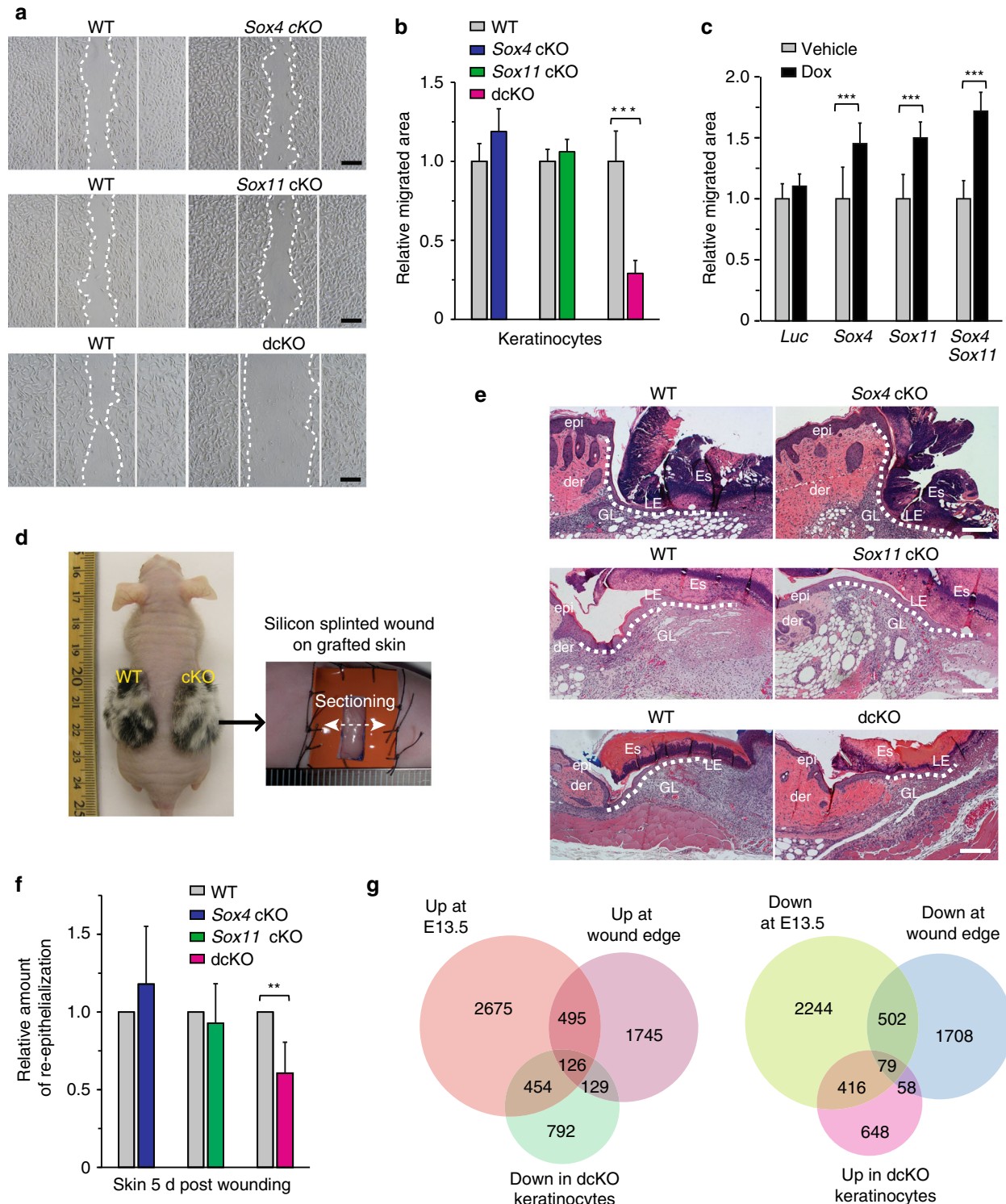

with EDC gene promoters, thus coordinating the concomitant expression of the long cluster of genes. Whether SOX11 and SOX4 exert their function in a long-range manner in this locus remains to be determined. Interestingly, our ChIP-seq data show that both SOX11 and SOX4 peaks are localized preferentially in enhancers enriched for the AP-1 binding motif. Since EDC genes are activated by the transcriptional activator AP-1[43,44] and repressed by SOX4 and SOX11, it is possible that in early development, SOX11 and SOX4 antagonize the recruitment of AP-1 to the EDC genes and/or abrogate AP-1's activity through a

direct interaction with the DNA-binding domain of AP-1[45], preventing EDC gene expression at that stage.

In addition to regulating the differentiation of epidermal cells, SOX11 and SOX4 also control migration, in line with their reported role in migration in mesenchymal stem cells, neuronal cells, and a variety of cancer cells[10,46–49]. Indeed, our transcriptomic data show that genes regulated by SOX11 and SOX4 in both embryonic and wounded epidermis are highly enriched in the GO categories related to cytoskeletal/ECM organization and cell motility.

**Fig. 8** Deficiency of *Sox11* and *Sox4* reduces cell migration and re-epithelialization. **a** Representative images of keratinocytes at the end point of the migration assay using primary keratinocytes from WT or cKO newborns. n = 3 biological replicates. **b** Graph quantifying relative areas the cells migrated normalized over control. Data are the mean ± SD. n = 3 biological replicates, with 16–24 fields quantified per replicate. ***p < 0.001 (Student's two-tailed *t*-test). **c** Effect of *Sox11* and *Sox4* on migration of cells lacking both genes. dcKO keratinocytes were transduced to express luciferase (*Luc*, control), *Sox11*, *Sox4*, or both *Sox11* and *Sox4*. Graph quantifying the relative areas the Dox-treated cells migrated normalized over vehicle controls. n = 2 biological replicates, with 12–20 fields quantified per replicate. Data are the mean ± SD. ***p < 0.001 (Student's two-tailed *t*-test). **d** Images of a nude mouse with grafted skins and the splinted wound healing model. WT and cKO neonatal dorsal skins were grafted onto *Nude* mice pairwise, and splinted wound was created ten weeks post grafting (4 × 10 mm). **e** Representative images of epidermal tongues five days post wounding in skin of indicated genotypes. **f** Graph quantifying the amount of re-epithelialization in cKO skin relative to littermate wild type controls. Data are the mean ± SD. n = 4 *Sox4* cKO-control pairs, 6 *Sox11* cKO-control pairs, and 6 dcKO-control pairs. **p < 0.01 (Student's two-tailed *t*-test). **g** Venn diagram (left) shows the overlapping of genes upregulated in epidermal cells at E13.5 and adult wounded edge[4], and downregulated in dcKO keratinocytes. The right diagram shows the overlapping of genes downregulated at the wound edge and at E13.5, and upregulated in dcKO keratinocytes. Epi, epidermis; Der, dermis; Es, eschar; LE, leading edge; GL, granulation layer; der, dermis. Scale bars, 100 μm (**a**), 200 μm (**e**). Source data for panels **b**, **c**, **f**, and **g** are provided as a Source Data file

---

**Table 1 . Embryonic and wounded epidermis signature genes that are directly regulated by SOX11 and SOX4**

Genes upregulated in epidermal cells at E13.5 and at wound edge that directly regulated by SOX11 and SOX4 in both E16.5 epidermis and keratinocytes
**Arhgef2**,**Etv4**,**Evl**,**Farp1**,**Fblim1**,**Fscn1**,Gm1673, Gng2,**Klhl22**,**Lgals1**, Limd2, Marcksl1, Pcolce,**Pxdn**, Rcor2,**Ror2**, Sccpdh,**Serpinf1**, Slc4A7, Snn, Tmcc2, TMEFF1, **Tmsb10**, Twist2,**Vcan**
Genes downregulated in epidermal cells at E13.5 and at wound edge that are directly regulated by SOX11 and SOX4 in both E16.5 epidermis and keratinocytes
Camta1, Chit1,**Rasgrp1**, Slco3a1

The table lists the genes found in both embryonic and wounded epidermis signature genes that are directly regulated by SOX11 and SOX4 in both keratinocytes and embryonic E16.5 epidermis. Genes related with cytoskeletal organization and motility are in bold font.

---

Amongst the embryonic genes induced at the wound edge, we identified *Fscn1* as a critical direct target of SOX11 and SOX4 regulating cell migration. In human, FSCN1 is low or negative in normal adult epithelia but abundant in embryonic stages, and upregulated in metastatic carcinoma[50]. FSCN1 functions as the primary actin cross-linker in filopodia and promotes cell migration and invasion[37,38,51,52]. Our work shows that FSCN1 expression is controlled by SOX11 and SOX4 during both development and wound repair.

Since ablation of *Fscn1* does not fully recapitulate the migration defect of the *Sox11*- and *Sox4*-double knockout keratinocytes and its induction does not fully rescue the dcKO cells, our findings strongly suggest SOX11 and SOX4 regulate migration through additional targets. Indeed in addition to *Fscn1*, our transcriptomic and ChIP-seq studies uncovered a number of SOX11- and SOX4 directly regulated genes implicated in cell adhesion/motility. Similarly to *Fscn1*, these genes are highly expressed in the embryonic stage and are upregulated at wound boundaries, suggesting a role in wound repair. The functional significance of these genes in cell migration and wound repair remains to be investigated.

Regulation of wound healing has been investigated in several developmental models across taxa, including embryos of *Drosophila*, *C. elegans*, chicken and mice. These studies showed that the embryonic epithelium uses almost the same cytoskeletal machineries during embryonic morphogenesis and wound repair[53,54]. Moreover, these molecular machineries are subjected to similar regulatory mechanisms during embryonic development and wound repair. In *Drosophila*, *grainy head* (*grh*, or *Grhl3* in mammals) and AP-1 function as major transcriptional regulators in epidermal development and wound repair. In mice, both are essential for skin morphogenesis and eyelid closure during development as well as wound repair[55–58]. GRHL3 and AP-1 regulate epidermal differentiation during morphogenesis and also facilitate cytoskeleton reorganization during the re-

epithelialization step of wound repair. Analogously, SOX11 and SOX4 play dual roles in both epidermal development and wound re-epithelialization by targeting genes involved in epidermal differentiation and cell organization/movement.

We should note that our study focused only on the re-epithelialization process, a crucial phase of the regeneration of the stratified epidermis, which takes place during first 3-10 days after wounding in our wound model. In cases when a large wound is created, new hair follicles are generated in the healing wound after 2 weeks, with the reactivation of many molecular components of embryonic follicle development during this regeneration process[59]. As we did not create large wounds nor follow the wound repair longer than 2 weeks, our experiments did not evaluate role of SOX11 and SOX4 in the regeneration of hair follicles after wounding. However, we suspect that they do not play an important role in this process, given that grafted skin deficient of *Sox11* and *Sox4* were competent to grow a full coat of hair. We postulate that the regeneration of the stratified epidermis and hair follicles after wounding requires different sets of genes, and that SOX11 and SOX4 are crucial for the regeneration of the stratified epidermis but not hair follicles following injury.

It remains to be determined whether SOX11 and SOX4 contribute to an altered chromatin state of cells at the wound edge and whether that is the mechanism underlying their induction of embryonic genes during wound repair. Moreover, how wounding triggers the induction of SOX11 in adult epidermis remains to be investigated. As our transcriptomic analysis clearly shows that epidermal morphogenesis and wound repair share many parallel molecular programs, we anticipate future studies will uncover that the induction of an embryonic program during tissue repair is a conserved phenomenon across multiple tissues.

## Methods
**Mice**. To generate conditional knockout mice, we intercrossed *Sox4*^fl/fl,*Sox11*^fl/fl; *Sox12*^−/− mice (129SvEx:C57BL/6 mixed background)[17,60] to *Krt14-Cre* mice

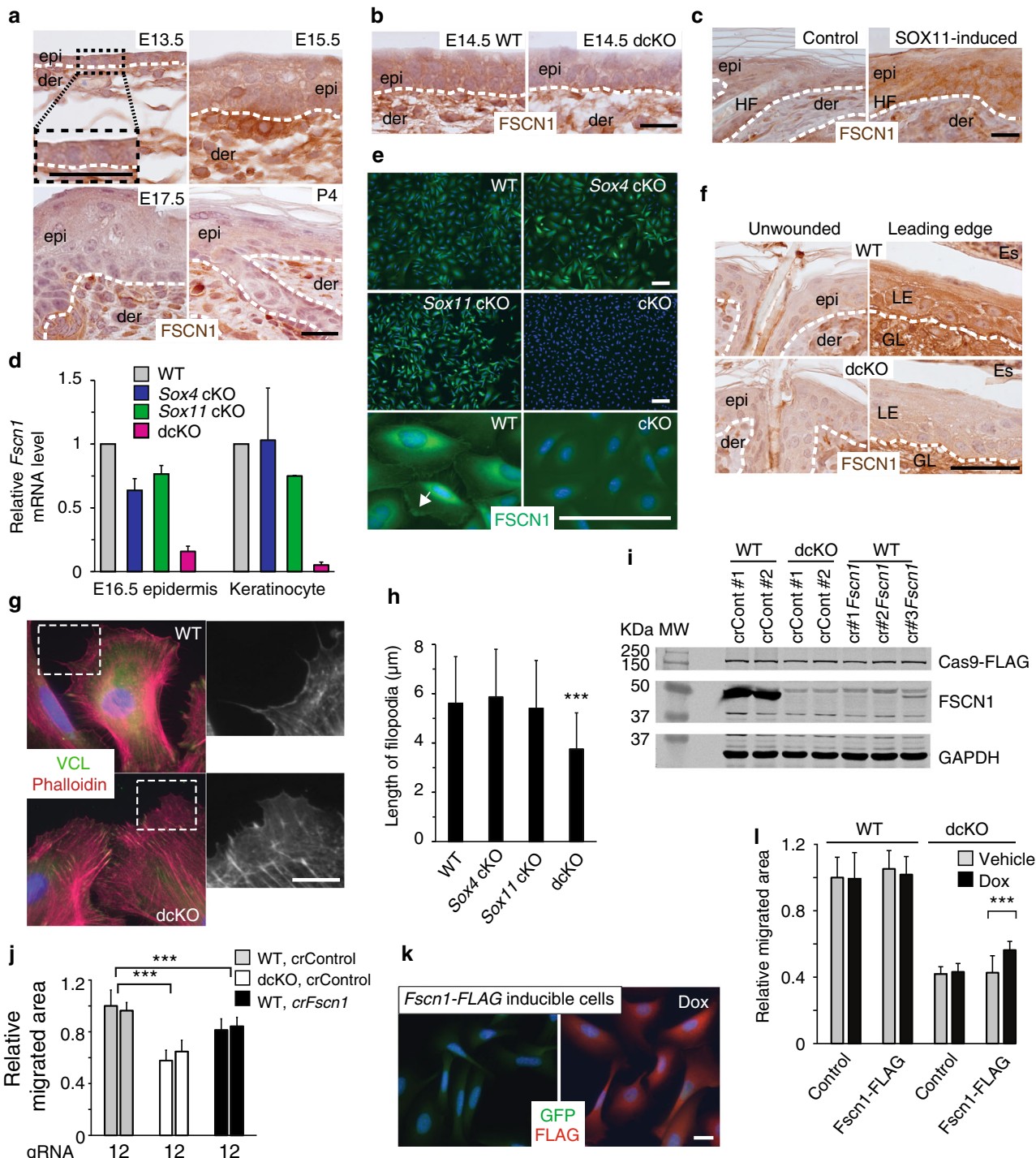

(FVB/N)[61]. We bred the heterozygous *Krt14-Cre;Sox4*fl/fl;*Sox11*fl/+; or *Krt14-cre;Sox4*fl/+;*Sox11*fl/fl mice to homozygous *Sox4*fl/fl;*Sox11*fl/fl mice to generate dcKO. Tet-inducible FLAG-epitope tagged *Sox11* transgenic mice were generated by crossing *Krt14*-rtTA (FVB/N)[23] with *TRE-Sox11* (FVB/N). The *Xho*I and *Drd*I fragment of p*TRE-Sox11* was microinjected into pronuclei of FVB/N embryos to generate the *TRE-Sox11* transgenic line. *Krt14-H2BGFP* transgenic mice (FVB/N) were obtained from the Fuchs Laboratory. Genotyping was confirmed by PCR with primers listed in Supplementary Table 4 or as reported[19,23].

For Dox-induction of *Sox11* transgene for less than 3 days, Dox (100ug; Sigma) were administered by i.p. injections. For induction beyond 3 days, mice were fed with doxycycline containing chow (625 mg/kg; Envigo). To induce SOX11-FLAG in grafted skin, dorsal skins from *K14-rtTA* (control) and *K14-rtTA/TRE-Sox11-FLAG* newborns were grafted in pair onto *Nude* mice, which were put on a diet containing Dox from post-grafting day 10 to day 28. For BrdU incorporation analysis in embryos, BrdU (50 μg/g of mouse; Sigma) was administered to pregnant mice by peritoneal injection and were sacrificed 4 h post injection.

Mice from both genders were used in experiments. All mice were maintained in the AALAC-accredited animal facility at Baylor College of Medicine and were used according to our protocol (AN-4907) approved by the Baylor College of Medicine institutional care and use committee.

**Barrier permeability and X-gal exclusion assay.** Mice were time-mated within a 10 h mating window and the mid-point of the mating window designated gestational age zero. X-gal exclusion assays were performed on embryos. Immediately after tails were snipped for genotyping, embryos were immersed in a low-pH X-gal substrate solution (1.3 mM MgCl$_2$, 3 mM K$_3$Fe(CN)$_6$, 3 mM K$_4$Fe(CN)$_6$, 1 mg/ml X-gal, 0.01% sodium deoxycholate, 0.02% NP-40, in 0.1 M citric acid/0.2 mM Na$_2$HPO$_4$, pH 4.5) at 32 °C for several hours to overnight until color develops. Tails were snipped to serve as a positive control for staining. At low pH abundant β-galactosidase in the skin cleaves X-gal, forming a blue precipitate in skin that are permeable to X-gal. The substrate can penetrate into skin until the epidermis has

**Fig. 9** *Fscn1* is regulated by SOX11 and SOX4 during development and wounding (**a**, **b**) Immunohistochemical analysis of FSCN1 expression in WT skin during development (**a**), in dcKO skin at E14.5 (**b**) and skin with FLAG-SOX11 induced since birth at P5 (**c**). **d** RT-qPCR analysis of *Fscn1* expression in E16.5 epidermis and primary keratinocytes with the indicated genotypes. Data are the mean ± SD. *n* = 3 biological replicates. **e** Immunofluorescence analysis of FSCN1 expression in keratinocytes of specified genotypes, with FSCN1 localized at cell edges (arrowed). **f** Immunohistochemical analysis of FSCN1 expression in WT and dcKO skin 5 days post wounding. **g** Immunofluorescence analysis of filopodia with anti-VCL (vinculin) and iFluor 647-conjugated phalloidin (to detect actin filaments). **h** Graph quantifying the filopodia length in keratinocytes. Data are the mean ± SD. *n* = 80 (WT), 63 (*Sox4* cKO), 83 (*Sox11* cKO), or 88 (dcKO). ***p < 0.001 (Student's two-tailed *t*-test). **i** Western analysis of WT or dcKO keratinocytes expressing FLAG-tagged CRISPR/Cas9 and control gRNAs or gRNAs targeting *Fscn1*. The representative blots from two independent experiments. **j** Graph quantifying relative areas the cells migrated normalized over the controls (WT cells expressing control gRNAs). *n* = 3 biological replicates, with 12–18 fields quantified per replicate. Data are the mean ± SD. ***p < 0.001 (Student's two-tailed *t*-test). **k** Immunofluorescence analysis of FLAG-FSCN1 expression in FACS-isolated GFP + cells from keratinocytes that were transduced to express GFP with tet-inducible FLAG-FSCN1 and with or without 24 h Dox treatment. **l** WT or dcKO keratinocytes were transduced to express GFP alone or GFP with FLAG-FSCN1. Graph quantifying relative areas the cells migrated normalized over the control (WT cells with empty vector without Dox). *n* = 3 independent experiments with 12–18 fields quantified per replicate. Data are the mean ± SD. ***p < 0.001 (Student's two-tailed *t*-test). Epi epidermis, der dermis, HF hair follicles, Es eschar; LE leading edge, GL granulation layer. Scale bars, 20 μm (**a–c**, **k**), 100 μm (**e**), 50 μm (**f**), 10 μm (**g**). Images are representative of 2 (**e**, **k**) or 3 (**a–c** and **f**) biological replicates. Source data for panels **d**, **h–j** and **l** are provided as a Source Data file

complete barrier function. After staining, embryos were embedded in agarose and photographed using a Zeiss SteREO Discovery.V8 microscope with transmitted and surface illumination.

**Skin grafting and splint wound healing assay.** The conditional double knockout mice (dcKO: *Krt14-Cre;Sox11*^fl/fl^;*Sox4*^fl/fl^;) could survive up to 12-16 h on the first postnatal day. The skin from sex-matched neonates was obtained shortly after the mice were born, and grafted pairwise onto the backs of nude mice (Taconic) after their genotypes were identified. After 6–8 weeks, skin grafts were subjected to splint full-thickness excisional wound healing assay with some modifications. After the nude mouse was anesthetized, the hair on the grafted skin was removed with clippers and then with depilatory cream. A mold was used to outline a 4 × 10-mm rectangular pattern for the wound where a full-thickness wound was then excised using an Iris scissor. A sterile silicon splint with a 6 × 12-mm hole was fixed to the skin around the wound with Krazy glue (Elmer's Inc.), and secured with interrupted sutures. The wounds were then dressed with self-adhesive elastic bandage. The skin-grafted mice were continuously checked for the intactness of the splints and the dressing.

To analyze re-epithelialization, the wounded skin was excised, bisected at the middle of the wound, and fixed in 10% formalin. The paraffin-embedded wound samples were sectioned (5-μm thickness) and stained with hematoxylin and eosin (H&E). Re-epithelialization was measured as the length of the extending epidermal tongue (EET) in micrometers.

**Primary keratinocyte isolation and culturing.** Primary keratinocytes were isolated through dispase and trypsin cell dissociation method[62]. Backskin from newborn pups were floated on PBS with dispase I (1 U/ml; Roche) at 4 °C overnight, allowing the epidermis to be peeled from the dermis the following day. The epidermis then were incubated in 0.05% trypsin for 5 min, neutralized with media containing FBS, and then pipetted up and down for 1 min, prior to being put through a 40 μ cell strainer. Keratinocytes were plated on J2 fibroblasts which were pretreated with Mitomycin C (8 μg/ml for 2 h). Cultured in E media with 15% serum 0.3 mM calcium, all keratinocytes were grown on feeder cells for three passages and off feeder cells at least two passages prior to being used in experiments.

**In vitro cell migration assay.** Confluent keratinocytes in 6-well plates were starved in serum-free basal medium with or without doxycycline for 22 h, followed by treatment with 10 μg/ml Mitomycin C for 2 h. After a scratch is made with a pipet tip across the center of the well, cells were washed three times with PBS to remove Mitomycin C and cell debris, and then were refed with serum-containing complete medium with or without doxycycline. Marks adjacent to the scratch were made as a reference for camera positioning. Cell migration was stopped 14–16 h later by fixing with 4% PFA. Photographs were taken at the initial and final time points using a phase-contrast microscope (Axiovert 40; Zeiss, Germany). By comparing the images from initial to final time points, the area filled by migrating cells was quantified with the ImageJ analysis software (http://rsb.info.nih.gov/ij/). Each experiment was done in replicates with at least two biological replicates and repeated at least twice. Values represent the mean ± SD of migrated area beyond the edges of the scratch.

**Epidermal cell isolation and FACS-purification.** Skins from *Krt14-H2BGFP* embryos at E13.5 were incubated in trypsin for 5 min at 37 °C, neutralized in media containing FBS, prior to being pipetted up/down 3–5 times and put through a 40 μ cell strainer. Dorsal skins from 4-day old pups (P4) were incubated with dispase I (1 U/ml) at 37 °C for 40–60 min and epidermal cells were isolated by the trypsin

method as described above. Dissociated cells were subjected to fluorescence-activated cell sorting (FACS), where cells were gated for single events and viability and sorted based on GFP fluorescence, yielding epidermal basal cells.

To isolate SOX11-induced epidermal cells, gender matched pairs of *Krt14-rtTA* and *K14-rtTA;TRE-Sox11* pups at P4 were injected with doxycycline for 12 h. Their dorsal skin was incubated with dispase I (1 U/ml) at 37 °C for 40–50 min to allow the separation of the epidermis from the dermis. The isolated epidermal sheets were promptly saved in TRIzol.

To obtain early E16.5 embryos, mice were time-mated within an 8 h mating window and the mid-point of the mating window designated gestational age zero. After the embryos were collected, the dorsal skin was incubated in dispase I (0.75 U/ml) at 37 °C for 30–40 min before the epidermis was peeled from the skin. To facilitate splitting the epidermis and preserve RNAs, skin was pretreated with RNALater. All the samples was stored in TRIzol in −80 °C prior to RNA extraction. The epidermal tissues were homogenized with beads (Precellys, Bertin Technologies) in 1-ml ice-cold TRIzol.

**Microarray sample preparation and analysis.** The quality of the total RNA samples was validated with the use of an Agilent 2100 Bioanalyzer, with the RNA integrity number (RIN) at 8 as the minimum acceptable value. The Genomic and RNA Profiling Core at Baylor College of Medicine followed the Agilent Two-Color Expression-Low Input Quick Amp protocol to process the samples as follows: 25 ng of total RNA from two biological replicates of each genotype, combined with RNA spike mix, were reverse transcribed using a T7 Primer Mix to produce cDNA. The cDNA product was transcribed using T7 RNA Polymerase, producing cyanine-3- and cyanine-5-labeled cRNA. The labeled cRNA was purified using a Qiagen RNeasy Mini Kit. A color swap was carried out to minimize dye specific effects. Purified products were quantified using the NanoDrop spectrophotometer for yield and dye incorporation, and tested for integrity on the Agilent Bioanalyzer. 300 ng of each labeled cRNA were fragmented. Approximately 480-ng total of each fragmented cRNA sample pair were loaded onto each of the Mouse G3 8 × 60K Expression arrays. The arrays were hybridized in an Agilent Hybridization Chamber for 17 h at 65 C with rotation at 10 rpm, and then washed using the Agilent Expression Wash Buffers One and Two, followed by acetonitrile. Once dry, the slides were scanned with the Agilent Scanner (G2565BA) using Scanner Version C and Scan Control software version A.8.5.1. Data extraction and quality assessment of the microarray data was completed using the Agilent Feature Extraction Software Version 11.0.1.1.

The Bioconductor package limma was used to analyze the microarray data. Briefly, the data were background corrected by the 'normexp' method with an offset of 16 added to the intensities. For each array, the R and G signal intensities were normalized by the loess method, which generated normalized M-values (log2 (R/G)) for each pair of samples. Principle component analysis was performed to check the color effect of cys5 and cys3 and the potential color effect was removed by using the "Combat" method. Moderated *t*-statistics were used to test if genes were differentially expressed between the groups of interest and Benjamini–Hochberg method was used to estimate false discovery rate (FDR). Probes with log2 fold change > 1.5 and FDR < 0.05 were considered differentially expressed.

Gene set enrichment analysis (GSEA) was carried out with 1000 permutations and other standard settings in the Broad institute GSEA software (http://software.broadinstitute.org/gsea/index.jsp). The embryonic epidermal signature gene set represents genes that are upregulated > 1.5 log2-fold change in E13 embryonic progenitors relative to the P4 epidermal basal cells (*n* = 2, FDR < 0.05).

**Constructs.** All PCRs were conducted using PrimeSTAR GXL DNA Polymerase (Clontech). To express of the intronless mouse *Sox4* and *Sox11* genes, their full-

length ORFs were cloned out from mouse embryonic stem cell genomic DNA. The PCR products were digested and ligated to pcDNA3.1 with 3× FLAG at the C-terminal. The 3× FLAG-tagged *Sox4* or *Sox11* in pcDNA3.1 vector was subcloned into *Eco*RI digested, blunted pENTR 1 A (Life Technologies) after *Eco*RV/*Pme*I digestion. To create myc-tagged *Sox4*, the *Sox4* ORF was PCR-amplified from its pcDNA3.1 vector with the addition of the *Hind*III and *Eco*RV sites with primers, and then ligated into pENTR 1 A with 3× myc at the C-terminal. *Fscn1* ORF was cloned from the first-strand cDNA from the mouse whole-brain tissue at E14.5.The pENTR-*LUC* (w158-1) was a gift from Eric Campeau (Addgene plasmid # 17473). To generate tet-inducible lentivirus constructs, the Gateway entry vectors were recombined into pINDUCER21 or pINDUCER20 vector using LR Clonase II (Life Technologies) according to the manufacturer's instructions.

Single guide RNA (sgRNA) target sequences were designed using the Benchling online tool (www.benchling.com) as 20-nt sequences preceding an NGG PAM in the genome (GRCm38). The oligo pairs were annealed and ligated into the *Bsm*BI-digested lentiCRISPR v2 vector[63] (Addgene plasmid # 52961).

*pTRE-Sox11* was engineered as follows: the β-globin intron was subcloned from the *Krt14* cassette vector[23] into the *Sac*I site of *pTRE2* (Clontech) to form the vector *pTRE2I*. The *Sox11-3×Flag* ORF from the *pcDNA3.1-Sox11-3×FLAG* expression vector was digested with *Xho*I and *Pme*I, then ligated into the *Sal*I/*Eco*RV-digested *pTRE2I*. pSUPER.retro vectors with shRNAs targeting *Fscn1* or *LacZ* (control) were a gift from Dr. Shengyu Yang from Penn State University[64].

To make reporter constructs, the SV40 promoter between *Bgl*II and *Hind*III in pGL3-promoter plasmid (Promega) was replaced with the TATA box, forming the pGL3-TATA vector. The enhancer or promoter fragments were amplified from mouse embryonic stem cell genomic DNA and ligated to the *Xho*I (*Sal*I compatible), *Bgl*II (*Bam*HI compatible), or *Xho*I/*Hind*III sites following digestion.

All plasmids were verified by restriction digestion and sequencing before use. All cloning primers are listed in Supplementary Table 3.

**Lentiviral production and transduction.** Lentiviral particles were generated in 293 T cells cultured in IMDM containing 10% FBS, 2 mM Supplementary ʟ-glutamine, and pen/strep. Cells were co-transfected with 7-µg pINDUCER vector and 750-ng each of pHDM-Hgpm2, pRc/CMV-RaII, pHDM-tat1b, and pHDM-VSV-G with 30 µL of TransIT-293 transfection reagent (Mirus Bio) per 10-cm plate, or with 3.6-µg lentiCRISPRv2 vector, 2.7-µg psPAX2, and 1.8-µg pMD2.G (Addgene plasmids 12259 and 12260) with 24 µL of TransIT-293 per 6-cm plate. Virus-containing supernatants were collected at 48 and 72 h post-transfection, pooled, and concentrated by PEG 8000 precipitation. Viral particles were resuspended in a small amount of keratinocyte growth media, separated into aliquots, and stored at −80 °C until use.

For lentiviral infections, keratinocytes were plated at $1 \times 10^5$ cells per well in 6-well tissue culture plates. The following day, one or two representative wells were trypsinized and counted. Lentivirus preps were thawed and diluted in keratinocyte growth media containing 8 µg/mL Polybrene (Sigma-Aldrich), to a final multiplicity of five transduction units per cell. Growth media was aspirated from the plated keratinocytes and replaced with diluted lentivirus (1 mL/well). The plates were incubated at 37 °C for 15 min and then centrifuged at 1100 g, 32 °C, for 30 min. After centrifuging, the cells were washed with 3 ml/well PBS, refed with keratinocyte growth media, and returned to 37 °C incubator. At 48 h post-infection, the cells were selected with puromycin (2 µg/ml final concentration), or passaged and cultured in 10-cm plates followed by FACS enrichment.

CRISPR/Cas9-treated keratinocytes were grown to 80–90% confluence and passaged at least three times before being used in experiments.

**RNA isolation and qPCR analysis.** All samples were stored in TRIzol in −80 °C prior to RNA isolation. Total RNA was purified with the PureLink RNA Mini kit (Life Technologies) and 1 µg of each RNA sample was reverse transcribed with the Superscript III First-Strand Synthesis System (Life Technologies) using Oligo(dT)$_{20}$ primers as recommended by the manufacturer. PCR amplifications of genes of interest were performed using primers located in different exons (or spanning intron-exon junctions) or in 3'UTR to obtain amplicons less than 150 bp in length. Real-time PCR was performed with Takyon No Rox SYBR MasterMix (Eurogentec) on a LightCycler 480 real-time PCR system (Roche). Differences between samples and controls were calculated based on the $2^{-\Delta\Delta CP}$ method. Relative cDNA copy numbers were determined with standard curves generated with genomic DNA templates. Expression of *Mrpl19* was used to normalize samples. Primer sequences are listed in Table S3.

**Immunoblotting.** Gel electrophoresis was performed using 30–40-µg protein extracted from TRIzol cell lysates resolved in 10–15% SDS-PAGE gels, semi-dry transferred for 2 h at 45 mA per unit to nitrocellulose membranes. Membranes were blocked for 1 h in 5% non-fat milk in 1× PBS, then incubated with primary antibodies in the BSA-TNTT solution (25 mM Tris-HCl, pH 8.0, 150 mM NaCl, 0.2% Tween-20, 2.5% BSA) overnight at 4 °C with gentle agitation. Membranes were rinsed 4× in TNTT buffer before incubating in secondary antibodies diluted in BSA-TNTT for 1 h at room temperature in the dark. Membranes were washed 4× in TNTT and transferred to 1× PBS. The blots were scanned on LI-COR Odyssey infrared imaging instrument. Antibodies were used at the following

concentrations: mouse anti-FLAG (1:1,500; Sigma, #F1804), rabbit anti-GAPDH (1:10,000; Bethyl, #A300-641A), mouse anti-FSCN1 (1:1,500; Santa Cruz, #sc-21743), and mouse anti-FBLIM1 (1:1,500; Santa Cruz, #sc-21743). All uncropped blots can be found in Supplementary Fig. 6 and in the Source Data file.

**Immunohistochemistry and immunofluorescence.** Frozen sections at 6–8 µm in thickness were cut on a Leica cryostat and stored in −80 °C after air-dried for 15–30 min. The slides were fixed in 4% paraformaldehyde (PFA) for 10 min, and blocked for 1 h with the MOM kit (Vector Laboratories, Burlingame, CA) for mouse monoclonal antibodies, or in the following blocking buffer for all other antibodies: 10% normal donkey serum, 2% BSA, 2% fish skin gelatin, 2% Triton X-100 in PBS. The sections then were incubated with primary antibodies diluted in the blocking buffer at 4 °C overnight at the following concentration: rabbit anti-SOX11 (1:200 for immunofluorescence; Sigma, #HPA000536), guinea pig anti-SOX11 antiserum (1:1,000 for immunohistochemistry; from Dr. Elisabeth Sock[21]), rat anti-BrdU (1:200; Abcam, #ab6326), rat anti-CD104 (1:200; BD Biosciences, #553745), rabbit anti-KRT5 (1:500; Covance, #PRB-160P), rabbit anti-KRT1 (1:500; Fuchs lab), rabbit anti-FLG (1:500; Covance, #PRB-417P), rabbit anti-LOR (1:500; Covance, #PRB-145P), rabbit anti-FLAG tag (1:500; Cell Signaling, #2368), guinea pig anti-TCF7L1 (1:100; Lab generated), rabbit anti-TCF7L2 (1:100; Cell Signaling, #2569), mouse anti-KRT18 (1:50; Cell Signaling Tech, #4546), mouse anti-FSCN1 (1:50; Santa Cruz, #sc-21743), mouse anti-FBLIM1 (1:100; Santa Cruz, #sc-21743), mouse anti-VCL (1:200; Novus Biologicals, # NB600-1293). Cyto-Painter Phalloidin-iFluor 647 Reagent (1:1,000; Abcam, #ab176759) was used to label actin filaments. As noted previously[65], no working antibody for SOX4 immunostaining is currently available.

Antigens were visualized with FITC- or RRX-conjugated secondary Abs (Jackson Labs). Hoechst 33342 dye was used to counterstain nuclei. For immunohistochemical detection, 5-µm paraffin sections were used; and endogenous biotin and peroxidase activity were blocked. The color was developed with DAB chromogen (ImmPACT DAB; VectorLabs, United States) and the sections were counterstained with hematoxylin. All images were acquired with Zeiss Axioskop microscope.

**Chromatin immunoprecipitation.** ChIP and ChIP coupled with massively parallel sequencing (ChIP-seq) were performed according to the established method[66] with some modifications as described below. Because no anti-SOX4 or anti-SOX11 antibodies were suitable for ChIP assay, we carried out ChIP assay with keratinocytes isolated from the dcKO newborns (*Krt14-Cre;Sox11^{fl/fl};Sox4^{fl/fl}*) that have been transduced with lentiviral vector pINDUCER21expressing FLAG-epitope tagged *Sox4* or *Sox11*. Keratinocytes were grown in 15-cm plates and treated with doxycycline 24 before fixation with 1% formaldehyde solution in 1×PBS for 10 min at room temperature. After the formaldehyde was quenched with 125 mM glycine, cells were rinsed 2× with 1×PBS and harvested using a cell scraper and flash frozen in liquid nitrogen. Cells were aliquoted and stored at −80 °C prior to use.

Nuclei were prepared from approximately 100 million cells and lysed in the sonication buffer (50 mM Tris-HCl pH 8, 10 mM EDTA, and 1% SDS). Chromatin was sheered by sonication in iced water (Bioruptor XL, Diagenode), 30 s on/30 s off, for 35 min, to achieve 100–500-bp DNA fragments. After dialysis against the low SDS buffer (20 mM Tris-HCl pH 8, 10 mM EDTA, and 0.01% SDS) overnight, the sonicated samples were centrifuged at 20,000g for 10 min. The soluble whole-cell extracts were added with Triton X-100 (final 1%) and NaCl (final 150 mM), and incubated with 100 µl of Dynal Protein G magnetic beads that had been pre-incubated with 10-µg mouse anti-FLAG antibody (Sigma) or mouse IgG. Beads were washed 1× with low-salt immune complex wash buffer (0.1% SDS, 1% TritonX-100, 2 mM EDTA, 150 mM NaCl, 20 mM Tris-HCl, pH8.1), 1× with high-salt immune complex wash buffer (0.1% SDS, 1% TritonX-100, 2 mM EDTA, 500 mM NaCl, 20 mM Tris-HCl, pH8.1), 1× with LiCl immune complex wash buffer (250 mM LiCl, 0.5% NP-40, 0.5% deoxycholate, 1 mM EDTA, 10 mM Tris-HCl, pH 8.1), and 1× with TE containing 50 mM NaCl. Bound complexes were eluted from the beads by heating at 65 °C for 30 min (50 mM Tris-HCl, pH 8.0, 10 mM EDTA, and 1% SDS) in a Thermomix. IgG ChIPs were performed under the same conditions as the experimental ChIPs. Crosslinking was reversed by incubating samples at 65 °C for 6 h. Whole-cell extract DNA reserved from the sonication step was treated in parallel to reverse crosslinking. After crosslink reversal, samples were treated with RNase and proteinase K prior to DNA extraction. ChIP-qPCR was performed before and after library construction with positive control primer sets for verification of enrichment.

For ChIP-qPCR analysis of epidermal cells isolated from Sox11-induced pups, 3-day old *K14-rtTA;TRE-Sox11-FLAG* pups were injected with 100 µl of 1 mg/ml Dox solution and their skins were harvested after 24 h. After treatment with neutral protease, basal cells were dissociated from the epidermis by trypsinization, before being pooled, counted, and fixed in 1% fresh formaldehyde. Epidermal cells freshly isolated from pups required harsher sonication condition than the condition used to process cultured keratinocytes for ChIP-seq. To dissociate the clumps and prepare nuclei, we forced the samples in Cell Lysis Buffer 1 (50 mM HEPES-KOH, pH 7.5, 140 mM NaCl, 1 mM EDTA, 10% glycerol, 0.5% NP-40, 0.25% Triton X-100) through 1 ml syringe attached with a 28G needle. 2–3 million cells in 130-µl sonication buffer were sonicated in microTUBE with AFA fiber with Covaris LE220 focused-ultrasonicator. Fragments of 200–900 bp were generated after 8–10-

min sonication. The fragments from ~10 million cells were pooled and dialyzed. The volume of the chromatin was brought to 1.5 ml with the ChIP SDS Dilution Buffer (0.01% SDS, 1% Triton X-100, 2 mM EDTA, 20 mM Tris-HCl, pH 8.1, 150 mM NaCl), and centrifuged at 20,000*g* for 10 min. The supernatant was pre-cleared with 30-μl Dynabeads pre-coated with BSA for 4 h before it was incubated with 6-μl FLAG antibody overnight. The antibody-chromatin complexes were retrieved with 30-μl Dynabeads, which were washed, followed by ChIPed DNA extraction and purification as described above.

**ChIP-seq sample preparation and analysis.** The Genomic and RNA Profiling Core at Baylor College of Medicine conducted sample quality checks using the NanoDrop spectrophotometer, Invitrogen Qubit 2.0 Fluorometer and Agilent Bioanalyzer 2100. The Rubicon ThruPlex DNA-Seq library preparation system (p/ n R400523, protocol QAM108-002) was used to prepare ChIP-Seq libraries for sequencing on the Illumina NGS sequencing instruments. A summary of the techniques and the data analysis are described below.

A double-stranded (ds) DNA libraries were generated from 0.5 ng of ChIP'd DNA of two biological replicates and were used for hybridization. This was achieved by first creating blunt ended fragments, then ligating stem-loop adapters with blocked 5' ends to the 5' end of the DNA, leaving a nick at the 3' end. Finally, library synthesis extends the 3' end of the DNA and Illumina-compatible indexes were incorporated with 11 amplification cycles. The fragments are purified using AMPure XP Bead Purification System, removing small sized fragments below 200 bp in size and unincorporated PCR reagents and nucleotides. The resulting libraries were quantitated using the NanoDrop ND-1000 spectrophotometer and fragment size assessed with the Agilent Bioanalyzer. A qPCR quantitation was performed on the libraries to determine the concentration of adapter-ligated fragments with the Applied Biosystems ViiA7 qPCR system and a KAPA Library Quant Kit.

dsDNA clusters were regenerated by bridge amplification. Using the concentration from the ViiA7 qPCR machine above, 16 pM of equimolarly pooled library was loaded onto a flowcell and amplified by bridge amplification using the Illumina HS2500 sequencing instrument. PhiX Control v3 adapter-ligated library was spiked-in at 5% to ensure balanced diversity and to monitor clustering and sequencing performance. A single-end 50 cycle run was used to sequence the flowcell on a HiSeq Sequencing System in Rapid Mode with v2 chemistry.

To analyze the ChIP-Seq data, ChIP-seq reads were mapped to the mouse genome (mm9) using Bowtie2 with default single-end settings[67]. Next, all non-nuclear, non-mapping, and low-quality reads were discarded. Duplicated reads were then removed with Picard MarkDuplicates. Peak calling was carried out with Homer (findPeaks -style factor)[68]. Blacklisted regions from mm9 were also removed from the individual and comprehensive peak files (bedtools –subtract). Motif enrichment analysis was conducted with Homer (findMotifsGenome.pl). All gene ontology (GO) enrichment analysis was performed using Metascape (http://metascape.org)[69].

**Reporter assay.** Keratinocytes were plated at density of 20,000 cells per well in a 24-well plate one day before being transfected with the empty control vector (pcDNA3) or vector expressing *Sox4* or *Sox11* along with the reporter constructs and the control vector pRL-TK, using TransIT-Keratinocyte Reagent (Mirus) according to the manufacturer's instructions. Luciferase reporter activity was measured using the Dual-Luciferase Reporter system 48 h after transfection (Promega) with *Firefly* luciferase values normalized to *Renilla* luciferase values.

**Statistical analysis.** Data were analyzed and statistics performed (unpaired or paired two-tailed student's *t*-tests). Error bars represent SD in all plots. Significance was set as *$p < 0.05$, **$p < 0.01$, and ***$p < 0.001$, unless otherwise indicated. Quantifications were performed from at least three independent experiments. In vivo re-epithelialization analysis was performed in a double-blinded fashion. Venn diagram hypergeometric $p$ values were calculated using R. Provided that in the $N$ total genes, the gene sets A (containing $m$ genes) and B (containing $n$ genes) have $k$ genes in the intersection, the enrichment level ($R$) of the overlap is defined as ($k \times N$)/($m \times n$).

**Reporting summary.** Further information on research design is available in the Nature Research Reporting Summary linked to this article.

## Data availability

Microarray and ChIP-seq data in this study have been deposited in the Gene Expression Omnibus (GEO), under accession codes: GSE120827 (transcriptional profiles of *Sox4* cKO, *Sox11* cKO, and *Sox4/11* dcKO mouse keratinocytes and their wild-type control), GSE120826 (transcriptional profiles of *Sox4* cKO, *Sox11* cKO, and *Sox4/11* dcKO mouse epidermis at E16), GSE120824 (transcriptional profiles of murine E13 epidermal cells and P4 epidermal basal cells), GSE120825 (transcriptional profile of SOX11-induced mouse epidermis), and GSE120773 (ChIP-seq), respectively. The source data underlying Figs. 1a–c, 3a, c, d, 4a, d–f, 5a–f, 6a, b, h, j, 7a, 8b, c, f, g, 9d, h–j, l and Supplementary Figs. 1c, 3, 4a–c and 5b–d are provided as a Source Data file. All data supporting the findings of this study are available upon request.

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

## Acknowledgements

The authors thank Drs. Geoffrey C. Gurtner and Kristine Rustad from Stanford University for their advice on the full-thickness splinted skin wound healing model; Dr. Shengyu Yang from Penn State University for insightful discussions on *Fscn1* studies; and Drs. Geneviera Allen from Rice University, Jun Wan from Indiana University, and Chih-Hsu Lin from Baylor College of Medicine for their suggestions on statistical data analysis. This project was supported in part by NCI and NIAMS (R21CA187368 and 2R56AR059122-06A1 to H.N.), with the technical support from the Genomic and RNA Profiling Core at Baylor College of Medicine with the assistance of Dr. Lisa D. White, Ph.D., and the Cytometry and Cell Sorting Core at Baylor College of Medicine, with funding from P30 Cancer Center Support Grant (NCI-CA125123).

## Author contributions

Q.M. and H.N. conceived the project and designed the experiments. Q.M., M.C.H., and H.N. wrote the manuscript. Q.M. performed most experiments. M.C.H. analyzed the ChIP-seq, deposited RNA-seq and microarray data for wounds, and gene ontology enrichment. F.J.C. and Q.X.M. performed microarray data analysis. A.K. and C.R. assisted Q.M. with some experiments. E.S. and V.L. provided essential reagents.

## Additional information

**Competing interests:** The authors declare no competing interests.

