## [Peer Review File · Nature Communications]

Reviewers' Comments:

Reviewer #1:

Remarks to the Author:

The manuscript by Miao et al describes the role of two transcription factors Sox11 and Sox4 in transcriptional regulation of an embryonic gene expression program in the epidermis that controls expression of late differentiation genes and genes regulating cell motility. This transcriptional program is re-initiated during wound healing in adult mice.

Overall this is an extensive and interesting study that sheds light to the poorly understood gene regulatory mechanisms of different transcriptional states and their plasticity in the epidermis. The study is in most part carefully done and the data is convincing. There are however a few issues that should be addressed prior to publication.

1. Transcriptome analysis of the Sox11, Sox4 and double KO mice: It is surprising that only knockout mice have been included in the analysis. To understand the functions of the Sox transcription factors it would be critical to include wild type mice in this analysis to set the proper baseline for the transcriptional profile of the epidermis at this developmental stage
2. ChIP experiment: Sox11 and Sox4 on expression of differentiation genes is to a large extent not retained in vitro in cell culture. In fact most of the differentiation genes are not increased in their expression (Fig S3). However, the ChIP seq is done from cultured cells. To what extent can the authors be confident that the genomic binding sites are retained in culture when the gene expression pattern is not preserved? Have they specifically looked at the promoters of genes that are differentially regulated in vivo vs in vitro? Are these perhaps not direct targets? If this is not the case then it should be emphasized that the regulation of differentiation is indirect.
3. The overexpressing FLAG transgenic mouse could be used to validate the ChIP data from freshly isolated cells.
4. A selected set of targets should be validated, for example by classical luciferase promoter activity assays.
5. Unlike the microarray data, the ChIP data has not been deposited in GEO allowing examination of the raw data. It is further not indicated how many biological replicates have been sequenced. It is also not clear what are the expression levels of the Flag-tagged proteins compared to the endogenous protein.
6. In general, a large amount of central data has been placed in the supplementary figures, whereas the main figures consist of very few panels. Prime examples of this are Figs 3 and 4 which contain very little data in comparison to the supplementary figures. Also, it is not clear why in the main Figure 4 a negative result on proliferation is shown, whereas key data on differentiation gene expression is buried in the supplementary. This negatively affects the readability of the paper. The authors should add as much data as possible in the main figures and only confirmatory or additional control experiments should be in the supplementary.

Reviewer #2:

Remarks to the Author:

In this study, authors showed that Sox11 and Sox4, transcription factors part of SOXC group, play distinct new role in skin epidermis of mice. Study conclusions' are drawn based on the large number of microarray expression data analyses and in vivo mutant mouse phenotypes. The overall amount of work is quite impressive. Focusing on Sox11, authors establish that its expression is high in early embryonic epidermis (E13.5), then rapidly decreases in late pre-natal and early post-natal times, stays turned-off in adult and then becomes re-activated in new epidermis forming over adult skin wounds. The key function of Sox11/4 in adult wound epidermis appears to be suppression of terminal differentiation program. I believe authors showed this quite conclusively. Additional role is to promote epidermal cell migration. Generally, increased proliferation, increased migration and temporarily suppressed terminal differentiation are the three hallmark features of

wound epidermis, and Sox11/4 play role in regulating two of these features. These reported roles of Sox11/4 in epidermis were previously not known.

However, I believe that authors over-interpret their findings and I do not believe they provided solid evidence for the two key claims in this manuscript: (1) that Sox11/4 reactivate embryonic epidermal program, and (2) that this is important for regeneration. Regeneration is defined as the ability of injured adult tissues to reactivate embryonic-like morphogenesis pathways, so that wounded tissues repair by restoring complex structures, whose formation is otherwise restricted to embryonic period. In mouse skin, such structures are hair follicles. They form embryonically, starting from around E13.5-14.5, but not in normal adult skin. Adult mouse skin is capable of embryonic-like regeneration of new hair follicles only in the center of large excisional wounds -- well-established phenomenon of Wound-Induced Hair Neogenesis (WIHN). However, this regeneration occurs only in very large wounds (more than 1cm² in size) and only in their center. Authors did not evaluate WIHN model but, instead, small wound model, where epidermis is known to not able to reactivate embryonic-like hair follicle regeneration. Small wounds repair by a substitute scar tissue and are covered by simple, flat epidermis. Considering this, authors' data does not permit to claim effects of Sox11/4 on regeneration.

Regarding their second key claim, authors did not provide sufficient evidence that Sox11/4 reactivate embryonic epidermal program. They microarray expression studies show that normal wound epidermis or Sox11-overexpressing epidermis signature genes partially overlap with E13.5 embryonic epidermis signature genes, but closer examination suggests that this overlap is actually not very large and is primarily limited to groups of genes involved in cell migration and terminal differentiation. Authors' own stringent analysis of genes shared by E13.5 epidermis, adult wound epidermis and suppressed upon Sox11/4 deletion in mutant epidermis (Fig. 6g) demonstrates that the number of shared genes is, in fact, very small. Furthermore, out of this shared small gene set, even fewer genes were validated to be direct targets of Sox11/4 on ChIP-seq (Fig. 6h). Embryonic epidermal program is characterized by many more properties than suppressed terminal differentiation and increased migration. Key feature of embryonic epidermis (particularly at E13.5) will be its morphogenetic competence to engage in skin appendage (hair follicle) development. I am not aware of the published data and author also did not show that normal wound epidermal cells from small wounds are capable of engaging in hair follicle development (such as in the context of hair reconstitution patch assays).

Therefore, in the context of provided data, authors need to significantly adjust their claims and primarily focus on the apparent new physiological role of Sox11/4 in transiently suppressing terminal differentiation in early wound epidermis.

Additional specific comments:

- 1) As a significant concern, biological replicate information is not provides for the most of the transgenic mouse experiments. I did not find it in the text nor in figure legends. This puts in question robustness of at least some reported observations.
- 2) In several Figures, quality of immunostaining data is not very high and there is substantial non-specific background. This includes Figs. 2e (KRT18), 7a, 7b, 7f, S6e and f.
- 3) Not all markers claimed on Fig. 2e as "early embryonic" are strictly embryonic. For example, TCF7L1 continues to be expressed in adult skin in hair follicle epithelial cells, including bulge stem cells and outer root sheath, and is also up-regulated in skin tumors (shown in earlier studies by the corresponding author group).
- 4) Results of Sox11 overexpressing skin grafts are very interesting and point toward a dramatic hair follicle defect (which remains to be addressed). Specific to epidermis -- at least based on histology (Fig. 2h), epidermis appears to be near normal and stratified. Is there time-dependent compensation for Sox11 over-expression and can authors evaluate differentiation marker

expression in these grafts to see if they become restored?

Reviewer #3:

Remarks to the Author:

The manuscript by Miao et al, "SOX11 and SOX4 drive the reactivation of an embryonic gene program during tissue regeneration," uses genomic interrogations of mouse genetic models to elucidate the role of SOX transcription factors in the regulation of epidermal tissue development and repair. The paper is well written and presents convincing insight into reactivation of embryonic gene regulatory networks that govern epidermal tissue regeneration. The experiments were executed thoroughly and carefully, and the data presentation is superb. The work makes substantial scientific impact and will be of interest to the fields of regenerative biology and gene regulatory networks.

Overall, I find that the paper is suitable for publication in Nature Communication. I offer few minor points the authors should address to improve the paper.

1. The title is a bit of an over-statement making it sound like they are presenting a broad interrogation of SOX11 and SOX4 in tissue regeneration. The title should be more tailored to the elucidated role for these SOX's specifically in epidermal wound repair.
2. P 3. Correct formatting of 'ref 19' (2 instances)
3. The study makes use of "wound-induced epidermal signature genes" from published data. It would be helpful to the reader to expound a bit (1-2 sentences in the introduction or at first mention in the Results) about the nature of that wounding model.
4. P 6. It would be a bit clearer to state that this analysis was performed on isolated keratinocytes (e.g. "...SOX11-FLAG is induced in keratinocytes isolated from tet-inducible...")
5. P6. "global transcriptomic changes" sounds a bit redundant. Perhaps better as just 'transcriptomic changes' or "global gene expression changes"
6. P 7, paragraph 1 (and also P9). The authors in mid paragraph shift their data analysis from "genes" to "probesets". And sometimes the results mention "genes" but figures (see 5d) show "probesets".
7. Fig 6 is missing panel C label, and has two D panels.

--- We appreciate the thorough and fair evaluation of our paper by the reviewers, who recognized the quality and innovation of our study. We really appreciate the reviewers' careful reading of our manuscript and providing us with very constructive suggestions. We were pleased that the reviewers appreciated our "*extensive and interesting study*" and found our work "*impressive*" and "*superb*". We were gratified that our findings were deemed to make "*substantial scientific impact and will be of interest to the fields of regenerative biology and gene regulatory networks*".

In our revised manuscript we addressed the concerns raised by the reviewers with additional data and clarification as advised. Outlined below is a summary of our revision.

- We revised the text to be more precise, replacing "tissue regeneration" with "wound healing".
- We optimized Keratin 18 and FSCN1 immunostaining condition to generate images with reduced background and provided additional immunostaining data as suggested.
- We also rearranged the figures, moving important supplementary figures to the main figures as recommended.
- To validate our ChIP-seq data derived from the SOX11-induced cultured keratinocytes as requested, we performed ChIP-PCR experiment on epidermal cells freshly isolated from SOX11-induced transgenic mice and used luciferase reporter gene assays to test the effect of SOX11/4 on selected target genes.

Figure 1: expanded to include original Fig. S1b-c

Figure 2: **improved K18 staining images (e), and new panels(i-j) showing differentiation markers in grafted SOX11-induced epidermis**

Figure 3: unchanged

Figure 4: expanded to include original Fig. S3d,e

Figure 5: original Fig 5a-d and Fig. S4a

Figure 6: original Fig. 5e-g and S4b-f with **New data (i) on luciferase reporter assays**

Figure 7: original Fig. 5h and S4g-i

Figure 8: original Fig. 6

Figure 9: original Fig. 7 and S6g-h, **improved FSCN1 staining images**

Sup Fig. 1: original Fig S2

Sup Fig. 2: original Fig S3

Sup Fig. 3: **New ChIP-PCR data on epidermal cells from SOX11-induced pups**

Sup Fig. 4: original Fig. S5

Sup Fig. 5: original Fig. S6a-f, **improved FSCN1 staining images**

We have highlighted the revised text in yellow. Below is our point-by-point response to the reviewers' comments.

Reviewer #1 (Remarks to the Author):

The manuscript by Miao et al describes the role of two transcription factors Sox11 and Sox4 in transcriptional regulation of an embryonic gene expression program in the epidermis that controls expression of late differentiation genes and genes regulating cell motility. This transcriptional program is re-initiated during wound healing in adult mice.

Overall this is an extensive and interesting study that sheds light to the poorly

understood gene regulatory mechanisms of different transcriptional states and their plasticity in the epidermis. The study is in most part carefully done and the data is convincing. There are however a few issues that should be addressed prior to publication.

1. Transcriptome analysis of the Sox11, Sox4 and double KO mice: It is surprising that only knockout mice have been included in the analysis. To understand the functions of the Sox transcription factors it would be critical to include wild type mice in this analysis to set the proper baseline for the transcriptional profile of the epidermis at this developmental stage

---We apologize for the confusion. We used Agilent two-color microarray, which applied values from wild-type littermate samples as the baseline. We have revised the text to better describe the experiment.

2. ChIP experiment: Sox11 and Sox4 on expression of differentiation genes is to a large extent not retained *in vitro* in cell culture. In fact most of the differentiation genes are not increased in their expression (Fig S3). However, the ChIP seq is done from cultured cells. To what extent can the authors be confident that the genomic binding sites are retained in culture when the gene expression pattern is not preserved? Have they specifically looked at the promoters of genes that are differentially regulated *in vivo* vs *in vitro*? Are these perhaps not direct targets? If this is not the case then it should be emphasized that the regulation of differentiation is indirect.

---We agree that unlike the *in vivo* situation, most EDC genes were not increased in dcKO keratinocytes. However, because gene expression is subject to other complex regulation other than transcription factor binding, including post-translational modification of the transcription factors, chromatin remodeling, as well as requirement of other cofactors, it is not surprising that many genes are differentially regulated *in vivo* and *in vitro*, given the different cellular contexts. The differential regulation doesn't necessarily mean the genomic binding sites identified *in vitro* are not preserved *in vivo*. Since transcription factors can bind to promoters/enhancers but might require other cofactors for its transactivation activity, lack of activity does not signify lack of binding nor does binding per se guarantee activity. Because we could not detect any obvious differences between promoters of genes differentially regulated *in vivo* vs *in vitro*, we think that it is the different cellular context that largely accounts for the differential regulation of the targets directly bound by SOX4/11.

3. The overexpressing FLAG transgenic mouse could be used to validate the ChIP data from freshly isolated cells.

---As suggested, we carried out ChIP-qPCR analysis of cells isolated from induced FLAG-SOX11 transgenic pups. We were able to validate the ChIP-seq targets that contain big SOX11-FLAG ChIP-seq peaks (≥ 15) but not those with smaller SOX11-FLAG ChIP-seq peaks (**Supplementary Fig. 3**). However, it should be noted that our condition of processing cells freshly isolated from postnatal skin significantly differed from the processing of cultured keratinocytes. We used 10 fold fewer cells with harsher

sonication condition to generate 200-900 bp fragments optimal for ChIP-qPCR. This may explain why our ChIP-qPCR analysis is less sensitive than the ChIP-seq, validating only ChIP-seq targets with high binding to SOX11-FLAG. We have added these caveats to our revised text.

4. A selected set of targets should be validated, for example by classical luciferase promoter activity assays.

---As suggested, we selected a few targets and validated the effect of SOX11 and SOX4 using classical luciferase promoter activity assays (**New Fig. 6i**). We cloned genomic regions containing the SOX11/4 ChIP-seq peak from the enhancer regions of *Tead2*, *Fscn1*, *Fblim1*, *Marcks11*, and *Pxdn* and placed them in either forward or reversed orientation upstream of the luciferase reporter gene. We found that SOX11 activates transcription of the reporter gene downstream of these sequences while SOX4 shows lower activities. This is not surprising since SOX4 has been found to have lower transactivating activity *in vitro* assays (Dy et al., 2008; doi: 10.1093/nar/gkn162).

5. Unlike the microarray data, the ChIP data has not been deposited in GEO allowing examination of the raw data. It is further not indicated how many biological replicates have been sequenced. It is also not clear what are the expression levels of the Flag-tagged proteins compared to the endogenous protein.

--We included the deposited raw ChIP data in GEO in the Reporting Summary in our submission and inadvertently did not include the link in the main text. Please see the link below.

https://genome.ucsc.edu/cgi-bin/hgTracks?hgS_doOtherUser=submit&hgS_otherUserName=mchill&hgS_otherUserSessionName=Qi_Miao_Sox4_Sox11_ChIPseq

We realized that we failed to include the number of biological replicates in several experiments as pointed out also by other reviewers. In the ChIP-seq experiment, we used 2 biological replicates for each transcription factor (SOX11 or SOX4). In the revised manuscript, we made sure that the number of biological replicates is specified in each experiment and included in the figure legends. Based on qPCR analysis of uninduced and induced SOX11-FLAG or SOX4-FLAG keratinocytes, the relative induced level is around 20 fold.

6. In general, a large amount of central data has been placed in the supplementary figures, whereas the main figures consist of very few panels. Prime examples of this are Figs 3 and 4 which contain very little data in comparison to the supplementary figures. Also, it is not clear why in the main Figure 4 a negative result on proliferation is shown, whereas key data on differentiation gene expression is buried in the supplementary. This negatively affects the readability of the paper. The authors should add as much data as possible in the main figures and only confirmatory or additional control experiments should be in the supplementary.

---We appreciate this very helpful suggestion. In order to include more supplementary figures in the main section, we have increased the number of main figures from 7 to 9. The rearrangement of the figures is described in the first section of our response.

--

Reviewer #2 (Remarks to the Author):

In this study, authors showed that Sox11 and Sox4, transcription factors part of SOXC group, play distinct new role in skin epidermis of mice. Study conclusions' are drawn based on the large number of microarray expression data analyses and in vivo mutant mouse phenotypes. The overall amount of work is quite impressive. Focusing on Sox11, authors establish that its expression is high in early embryonic epidermis (E13.5), then rapidly decreases in late pre-natal and early post-natal times, stays turned-off in adult and then becomes re-activated in new epidermis forming over adult skin wounds. The key function of Sox11/4 in adult wound epidermis appears to be suppression of terminal differentiation program. I believe authors showed this quite conclusively. Additional role is to promote epidermal cell migration. Generally, increased proliferation, increased migration and temporarily suppressed terminal differentiation are the three hallmark features of wound epidermis, and Sox11/4 play role in regulating two of these features. These reported roles of Sox11/4 in epidermis were previously not known.

However, I believe that authors over-interpret their findings and I do not believe they provided solid evidence for the two key claims in this manuscript: (1) that Sox11/4 reactivate embryonic epidermal program, and (2) that this is important for regeneration. Regeneration is defined as the ability of injured adult tissues to reactivate embryonic-like morphogenesis pathways, so that wounded tissues repair by restoring complex structures, whose formation is otherwise restricted to embryonic period. In mouse skin, such structures are hair follicles. They form embryonically, starting from around E13.5-14.5, but not in normal adult skin. Adult mouse skin is capable of embryonic-like regeneration of new hair follicles only in the center of large excisional wounds -- well-established phenomenon of Wound-Induced Hair Neogenesis (WIHN). However, this regeneration occurs only in very large wounds (more than 1cm² in size) and only in their center. Authors did not evaluate WIHN model but, instead, small wound model, where epidermis is known to not able to reactivate embryonic-like hair follicle regeneration. Small wounds repair by a substitute scar tissue and are covered by simple, flat epidermis. Considering this, authors' data does not permit to claim effects of Sox11/4 on regeneration.

Regarding their second key claim, authors did not provide sufficient evidence that Sox11/4 reactivate embryonic epidermal program. They microarray expression studies show that normal wound epidermis or Sox11-overexpressing epidermis signature genes partially overlap with E13.5 embryonic epidermis signature genes, but closer examination suggests that this overlap is actually not very large and is primarily limited to groups of genes involved in cell migration and terminal differentiation. Authors' own stringent analysis of genes shared by E13.5 epidermis, adult wound epidermis and suppressed upon Sox11/4 deletion in mutant epidermis (Fig. 6g) demonstrates that the number of shared genes is, in fact, very small. Furthermore, out of this shared small gene set, even fewer genes were validated to be direct targets of Sox11/4 on ChIP-seq (Fig. 6h). Embryonic epidermal program is characterized by many more properties than suppressed terminal differentiation and increased migration. Key feature of embryonic

epidermis (particularly at E13.5) will be its morphogenetic competence to engage in skin appendage (hair follicle) development. I am not aware of the published data and author also did not show that normal wound epidermal cells from small wounds are capable of engaging in hair follicle development (such as in the context of hair reconstitution patch assays).

Therefore, in the context of provided data, authors need to significantly adjust their claims and primarily focus on the apparent new physiological role of Sox11/4 in transiently suppressing terminal differentiation in early wound epidermis.

---We used the term regeneration to mean regeneration of the stratified epidermis and did not intend to mean regeneration of the epidermis and its appendages. We regret this choice of wording and have revised our text accordingly, replacing the term regeneration with wound repair.

---Our claim that Sox11/4 activate the embryonic epidermal program was based on the comparison of Sox11-induced and Sox11/4 deficient epidermal transcriptomes with embryonic epidermal signature genes. Close to 50% of genes increased by Sox11 overexpression are genes enriched in embryonic epidermis while 50% of genes downregulated by Sox11 were genes repressed in embryonic epidermal signature (**Fig. 3d**). Conversely, 60% of genes increased in *Sox11/4* dCKO were genes repressed in embryonic epidermis while close to 70% of genes downregulated in *Sox11/4* dCKO were genes enriched in embryonic epidermis (**Fig. 5f**). In addition, when we incorporated the ChIP-seq data with transcriptomic data, we found that 60% of direct targets of SOX11/4 are part of the embryonic signature genes (**Fig. 7a**).

---Since more than half of the genes altered by Sox11/4 expression are part of the embryonic signature genes, we postulate that Sox11/4 activates the embryonic epidermal program. However, we did not intend to claim that the embryonic epidermal program is controlled solely by Sox11/4, but only to point out that a significant number of downstream targets of Sox11/4 are part of the embryonic epidermal program.

We agree with the reviewer that “Embryonic epidermal program is characterized by many more properties than suppressed terminal differentiation and increased migration”. It is possible that SOX11 and SOX4 regulate primarily these aspects of the embryonic epidermal program and other regulators control other parts of the program. That a large number of genes regulated by SOX11 and SOX4 are involved in differentiation and migration is in fact consistent with the observed phenotype of mice overexpressing Sox11 and deficient of Sox11/4.

---We understand the reviewer’s concern of the low number of epidermal embryonic genes induced at the wound edge that are regulated by SOX11 and SOX4 in the original Fig. 6g (New 8g); however, we think that SOX11/4 controlling the expression of 126 of 621 embryonic genes induced at the wound edge is an impressive number (20% of the total).

---As for the analysis in the original Fig. 6h (New 8h) where we examined genes directly regulated by SOX11/4 identified in both keratinocytes and E16.5 epidermal cells, we specifically used this stringent setting to identify genes that are likely to function *in vivo* as well as *in vitro*. We are aware that this stringent analysis eliminates downstream indirect targets as well as genes that SOX11/4 control *in vivo* and not *in vitro* and vice versa. We therefore are not concerned about the low number of genes identified, since

our analysis achieved our goal of identifying a small set of target genes that we could analyze to better understand how SOX11/4 act to control migration and wound repair.

Additional specific comments:

1) As a significant concern, biological replicate information is not provided for the most of the transgenic mouse experiments. I did not find it in the text nor in figure legends. This puts in question robustness of at least some reported observations.

---We thank the reviewer for pointing out this important oversight. We made sure that the revised figure legends now include the number of biological replicates used in each experiment.

2) In several Figures, quality of immunostaining data is not very high and there is substantial non-specific background. This includes Figs. 2e (KRT18), 7a, 7b, 7f, S6e and f.

---We agree that the images of KRT8 and FSCN1 immunostaining showed high non specific background, which resulted from the use of mouse antibodies on mouse tissues. We succeeded in reducing the background by using a different antibody against KRT8 and changing FSCN1 staining condition. We have replaced the old images with the new ones in the revised figures (**New Fig 2e, Fig. 9a-c, f, Supplementary Fig. 5e-f**).

3) Not all markers claimed on Fig. 2e as "early embryonic" are strictly embryonic. For example, TCF7L1 continues to be expressed in adult skin in hair follicle epithelial cells, including bulge stem cells and outer root sheath, and is also up-regulated in skin tumors (shown in earlier studies by the corresponding author group).

---We should have been more precise in calling TCF7L1 an embryonic marker. TCF7L1 is indeed abundantly expressed in hair follicle stem cells, hair germs, and the outer root sheath, as we have shown in our previous studies. We referred to TCF7L1 as an embryonic marker because of its expression in early embryonic but not postnatal interfollicular epidermis. We have clarified the text accordingly to be more accurate.

4) Results of Sox11 overexpressing skin grafts are very interesting and point toward a dramatic hair follicle defect (which remains to be addressed). Specific to epidermis -- at least based on histology (Fig. 2h), epidermis appears to be near normal and stratified. Is there time-dependent compensation for Sox11 over-expression and can authors evaluate differentiation marker expression in these grafts to see if they become restored?

---Indeed, overexpressing Sox11 prevents hair follicle development. However, we did not investigate this any further, as it was beyond the scope of our focus, which is to define the role of Sox11 in the differentiation and repair of the stratified epidermis. Despite the near normal looking histology of the grafted epidermis in Fig. 2h., immunostaining analysis of grafted skin clearly shows drastically reduced expression of differentiation markers Loricrin and Filaggrin (**Fig. 2i-j**).

--

Reviewer #3 (Remarks to the Author):

The manuscript by Miao et al, "SOX11 and SOX4 drive the reactivation of an embryonic gene program during tissue regeneration," uses genomic interrogations of mouse genetic models to elucidate the role of SOX transcription factors in the regulation of epidermal tissue development and repair. The paper is well written and presents convincing insight into reactivation of embryonic gene regulatory networks that govern epidermal tissue regeneration. The experiments were executed thoroughly and carefully, and the data presentation is superb. The work makes substantial scientific impact and will be of interest to the fields of regenerative biology and gene regulatory networks.

Overall, I find that the paper is suitable for publication in Nature Communication. I offer few minor points the authors should address to improve the paper.

---We thank the reviewer for very constructive suggestions, which we fully addressed as described below.

1. The title is a bit of an over-statement making it sound like they are presenting a broad interrogation of SOX11 and SOX4 in tissue regeneration. The title should be more tailored to the elucidated role for these SOX's specifically in epidermal wound repair.

---We used the term "regeneration" to mean regeneration of the stratified epidermis, not realizing that this could be misleading. We have revised the text to be more precise.

2. P 3. Correct formatting of 'ref 19' (2 instances)

---We used Sox4 (ref 19) instead of Sox4¹⁹, because we were under the impression that the superscript number should not be used after a number. We fixed the formatting as suggested but will defer to the editor on the choice of formatting.

3. The study makes use of "wound-induced epidermal signature genes" from published data. It would be helpful to the reader to expound a bit (1-2 sentences in the introduction or at first mention in the Results) about the nature of that wounding model.

---As suggested, we briefly described how the published "wound-induced epidermal signature genes" were generated. We added the following sentence in the first mention in the results: "The wound-induced epidermal signature genes were generated from comparative RNA-seq analysis of hair follicle stem cells (HFSCs) from unwounded skin and HFSCs and their progeny around the biopsied-wounded area 7 days post wounding."

4. P 6. It would be a bit clearer to state that this analysis was performed on isolated keratinocytes (e.g. "...SOX11-FLAG is induced in keratinocytes isolated from tet-inducible...")

---We did not use the term keratinocytes, because we used the entire stratified epidermis isolated from dispase-treated skin, which likely contain some embedded melanocytes as

well as merkel and Langerhans cells. This is in contrast to the E13.5 and P4 purified basal keratinocytes, which were FACS-isolated from K14-GFP pups based on GFP fluorescence.

5. P6. "global transcriptomic changes" sounds a bit redundant. Perhaps better as just "transcriptomic changes" or "global gene expression changes"

---We have revised the text as suggested.

6. P 7, paragraph 1 (and also P9). The authors in mid paragraph shift their data analysis from "genes" to "probesets". And sometimes the results mention "genes" but figures (see 5d) show "probesets".

---Although the analysis of number of genes or probesets do not change the result outcome, we agree with the reviewer that it would be better to present consistent analysis. We have revised the figures to show analysis of the number of genes instead of probesets in Fig 3d and Fig. 5f (original 5d). In other sets of data, we used probesets to compare data from the same platform (Agilent microarray) and genes when comparing microarray data to RNA-seq or ChIP-seq.

7. Fig 6 is missing panel C label, and has two D panels.

---Thank you for pointing out this labeling error, which is now corrected.

Reviewers' Comments:

Reviewer #1:

Remarks to the Author:

The authors have addressed all my concerns and the manuscript has improved significantly

Reviewer #2:

Remarks to the Author:

This is an exciting manuscript, that was significantly improved following the revision. In principle, I believe it is suitable for publication.

Authors appropriately changed terminology in the Title and Abstract from "regeneration" to "wound repair." However, since throughout the text authors continue to refer to embryonic gene program and (in the Introduction) they discuss the pro-regenerative function of SOX genes in other organs, it is also appropriate to mention that adult skin can be induced to undergo embryonic-like regeneration by wounding (PMIDs: 17507982, 28059714). In the Discussion, authors can speculate if SOX genes are also involved in such, more profound regenerative response by the skin, that goes beyond re-epithelialization.

Reviewer #3:

None

REVIEWERS' COMMENTS:

Reviewer #1 (Remarks to the Author):

The authors have addressed all my concerns and the manuscript has improved significantly

Reviewer #2 (Remarks to the Author):

This is an exciting manuscript, that was significantly improved following the revision. In principle, I believe it is suitable for publication.

Authors appropriately changed terminology in the Title and Abstract from "regeneration" to "wound repair." However, since throughout the text authors continue to refer to embryonic gene program and (in the Introduction) they discuss the pro-regenerative function of SOX genes in other organs, it is also appropriate to mention that adult skin can be induced to undergo embryonic-like regeneration by wounding (PMIDs: 17507982, 28059714). In the Discussion, authors can speculate if SOX genes are also involved in such, more profound regenerative response by the skin, that goes beyond re-epithelialization.

Please note that Referee #3 provided comments to the Editor: "I am satisfied with the authors responses and associated revisions to my comments."

RESPONSE

-----We are gratified that our revised manuscript has satisfied the concerns of all reviewers. Reviewer #2 has asked us to mention the regeneration of hair follicles after wounding in adult skin and to speculate on the role of SOX11 and SOX4 in this process. We have added the following section in the discussion as suggested.

"We should note that our study focused only on the re-epithelialization process, a crucial phase of the regeneration of the stratified epidermis, which takes place during first 3-10 days after wounding in our wound model. In cases when a large wound is created, new hair follicles are generated in the healing wound after two weeks, with the reactivation of many molecular components of embryonic follicle development during this regeneration process⁶⁰. As we did not create large wounds nor follow the wound repair longer than 2 weeks, our experiments did not evaluate role of SOX11 and SOX4 in the regeneration of hair follicles after wounding. However, we suspect that they do not play an important role in this process, given that grafted skin deficient of *Sox11* and *Sox4* were competent to grow a full coat of hair. We postulate that the regeneration of the stratified epidermis and hair follicles after wounding requires different sets of genes, and that SOX11 and SOX4 are crucial for the regeneration of the stratified epidermis but not hair follicles following injury."